# VISUAL EVIDENCE PROMPTING MITIGATES HALLUCINATIONS IN LARGE VISION-LANGUAGE MODELS

## ABSTRACT

Despite the promising progress achieved, Large Vision-Language Models (LVLMs) still suffer from the hallucination problem, *i.e.*, they tend to predict objects and relations which are non-existent in the target images. These unfaithful outputs degrade the model performance and greatly harm the user experiences in real-world applications. Fortunately, traditional small visual models excel at producing professional and faithful outputs, but they are not adept at interacting with humans. Therefore, this work explores how small visual models complement the LVLMs by effectively extracting contextual information from images to generate precise answers. In particular, we show how such hallucination mitigates naturally in LVLMs via a simple method called visual evidence prompting, where a few visual knowledge evidences are provided as contexts in prompting. Experiments on three large language models show that visual evidence prompting improves performance on the evaluation of object hallucinations, as well as the new benchmark for relation hallucinations. We hope our work will not only serve as the minimal strongest baseline for the challenging hallucination benchmarks, but also highlight the importance of carefully exploring and analyzing the enormous visual evidence hidden inside small visual models before crafting finetuning LVLMs.

## 1 INTRODUCTION

The success of large vision-language models (LVLM) has resulted in significant advancements in overall comprehension of visual semantics (Chen et al., 2023b; Li et al., 2023a). Despite the success, it also introduces a notable issue being their tendency to produce hallucinations. They tend to produce non-existent objects (there is a "chair" in the image) and relations (dog is "behind" the cup) in the image (Li et al., 2023b; Gunjal et al., 2023; Liu et al., 2023a). Addressing and mitigating these hallucinations is crucial to improve the reliability and accuracy of vision-language models in real-life use cases.

Detecting and dealing with phenomenon proves to be challenging and often needs human supervision. Prior approaches has given models the ability to generate faithful responses by annotating negative instructions or unfaithful object descriptions and relations (Gunjal et al., 2023; Liu et al., 2023a). It is costly to create a large set of high quality answers, which is much more complicated than simple input–output pairs used in normal machine learning. More importantly, during instruction tuning of large vision-language models, there is a risk of overly optimizing the model to fit a specific problem or dataset. This tuning approach may lack generalization ability and lead to catastrophic forgetting, making it incompatible with other models or problems Zhai et al. (2023).

Fortunately, traditional small visual models excel at the tasks they are trained for. For instance, in the task of object detection, small visual models can efficiently identify and locate objects within an image (Fang et al., 2021; Carion et al., 2020). In the task of scene graph generation (SGG) (Zellers et al., 2018; Cong et al., 2023), small visual models can generate detailed descriptions of objects and their relations within a given scene, such as "a person sitting on a chair" or "a car parked next to a building". Small visual models are better characterized as narrow experts who focus on the processing and understanding of visual content, while LVLMs are competent generalists who have strong semantic understanding and generalization capabilities. Naturally, the small visual models complement the LVLMs by effectively extracting contextual information from images to generate

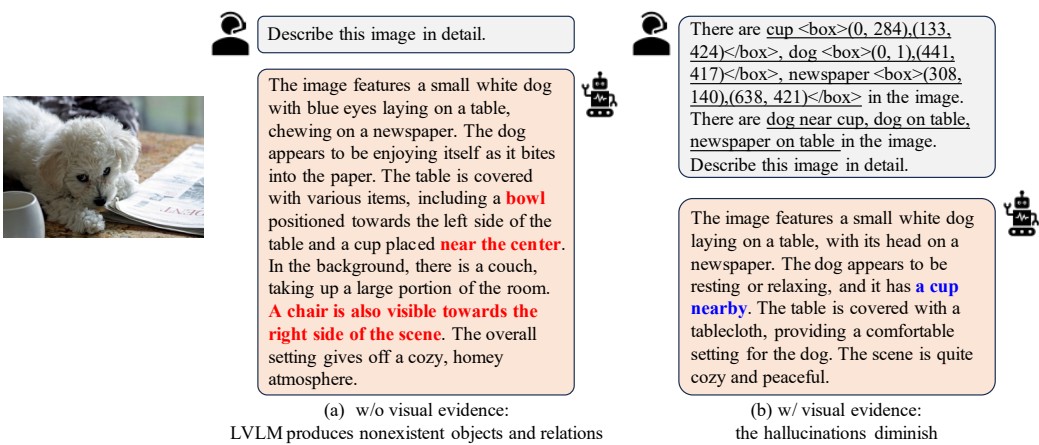

Figure 1: Hallucination in LVLM (QWen-VL-Chat). (a): LVLM produces **non-existent** objects (chair, bowl) and relations (cup near the center). (b): small visual models (object detection and scene graph generation) accurately output the object (cup, dog, newspaper) and relations (dog near cup). With the output of visual models as contexts in prompting, the fictional chair and bowl diminishes in the answer, and the relation between dog and cup is **precisely** described referred to the evidence.

precise answers. Moreover, due to the limited semantic understanding, small visual models are nearly incapable of producing illusions, which further enhances their ability to complement larger models.

This work explores how the hallucinations of LVLMs can be mitigated by referring to visual evidence from small visual models. An example is shown in Figure 1. The original LVLM produces the non-existent objects ("chair", "bowl") and relations ("cup near the center") in the answer. At the same time, small visual models, *i.e.*, object detection and scene graph generation models, output accurate objects and relations, *e.g.*, "cup $\langle 0, 284, 133, 424 \rangle$", "dog near cup". Symbolizing the accurate and faithful output of visual models as context prompts, the non-existent chair and bowl diminished in the final answer and the relation between dog and cup is precisely described referred to the evidence. We refer to this approach as *visual evidence prompting* during the inference.

To the best of our knowledge, we the first to study the visual evidence. Our method is partially inspired by the knowledge evidence solutions to tackle the hallucination in language models. These approaches retrieve supporting *evidence* from the knowledge base or encyclopedia to corrects factual errors in the output. While the visual knowledge base is infeasible in the vision-language tasks, the output of visual models is analogous to the knowledge evidence. The difference lies in that the evidence for large language models is static and consists of common knowledge, whereas the evidence for vision-language models is dynamic and associated with the content of specific images.

To evaluate the hallucinations of object and relation together, we introduce a new dataset and benchmark for the relation hallucinations. Our empirical evaluations on two benchmarks show that visual evidence prompting outperforms standard prompting, sometimes to a striking degree, as well as the instruction tuning methods. The evaluations also show that current LVLMs are more prone to encounter relation hallucinations, and our method exhibits greater gains in addressing relation hallucinations. Furthermore, we also conduct in-depth analysis about the robustness against visual models capacities, erroneous visual evidence, prompt templates and image domains. We aim for our work to not only establish a minimal yet robust baseline for the challenging benchmarks, but also draw attention to the crucial need for thorough exploration and analysis of the vast visual evidence concealed within small visual models, prior to crafting and fine-tuning LVLMs.

## 2 RELATED WORKS

### 2.1 LARGE VISION-LANGUAGE MODELS

Large vision-language models have seen performative advancements in tasks such as generating text from images and multi-modal in-context learning (Chen et al., 2023b; Li et al., 2023a). Recent

work has focused on utilizing instruction tuning techniques to enhance the zero-shot performance of instruction-aware LVLMs across different vision-language tasks Liu et al. (2023c); Dai et al. (2023). Specifically, LLaVA Liu et al. (2023c) projects the output of a visual encoder as input to LLaMA Touvron et al. (2023) and trains both the alignment network and the LLM on synthetic data. MiniGPT4 Zhu et al. (2023) is built on BLIP-2 Li et al. (2023a) but uses Vicuna Platzer & Puschner (2021) as the language decoder. MiniGPT-4 aims to align visual information from a pretrained vision encoder with an advanced large language model (LLM). mPLUG-Owl (Ye et al., 2023) incorporates a visual abstractor to bridge pretrained visual encoder ViT-L/14 and LLM (LLaMA) with a two stage finetuning procedure. Qwen-VL-Chat consists of two stages of pre-training and a final stage of instruction tuning training.

## 2.2 HALLUCINATIONS IN LARGE LANGUAGE MODELS

The extraordinary capabilities of LLMs come with a significant drawback: their potential to generate unsupported text due to their lack of understanding of what is factual and what is not Maynez et al. (2020); Krishna et al. (2021); Longpre et al. (2021). As a result, there has been a surge of interest to address LLM hallucination through knowledge-grounded neural language generation. To address this limitation, various works augment LLMs with knowledge consisting of personalized recommendations Ghazvininejad et al. (2017), Wikipedia article and web search Dinan et al. (2018); Shuster et al. (2022), structured and unstructured knowledge of task-oriented dialog Peng et al. (2022). In the LVLMs, it is infeasible to acquire grounded knowledge from a general knowledge base.

## 2.3 HALLUCINATIONS IN LARGE VISION-LANGUAGE MODELS

Despite the success of LVLMs, previous work has revealed that both LLMs and LVLMs suffer from hallucination. Similar to LLMs, LVLMs tend to generate non-existent objects in a target image. In the literature of computer vision field Rohrbach et al. (2018); Biten et al. (2021). object hallucination refers that the model generates descriptions or captions that contain objects which are inconsistent with or even absent from the target image. In general, object hallucination can be defined at different semantic levels. The most straightforward way is to define it over the object level. More fine-grained definitions might be concerned with the relations of objects. In this work, we focus on coarse-grained object hallucinations and fine-grained relation hallucinations at the same time, other hallucinations such as the number and attributes of the object are left for future work. In previous works Li et al. (2023b), the evaluation metric "POPE" is proposed to evaluate hallucinations in LVLMs by polling questions about generated text. They observed that current state-of-the-art LVLM (InstructBLIP Dai et al. (2023)) has the lowest object hallucination rates among recent LVLMs. Gunjal et al. (2023) created a hallucination dataset and optimized the InstructBLIP over the dataset with variation of Direct Preference Optimization Rafailov et al. (2023). These studies collectively contribute to the understanding and mitigation of hallucination-related challenges in LVLMs, by providing evaluation metrics, datasets, and tuning methods that enhance the reliability and consistency of the generated answers. Yet, there is a risk of overly optimizing the model to fit a specific problem or dataset, leading to catastrophic forgetting and lack of generalization ability Zhai et al. (2023).

## 3 VISUAL EVIDENCE PROMPTING

The goal of this paper is to endow large vision-language models with the ability to mitigate the hallucinations with the help of visual evidence. Generating answer with an input image and query question can be expressed in a probabilistic framework as estimating a conditional distribution $p(answer|question, image)$. The visual evidence prompting is formalized as $p(answer|question, image, evidence)$, where the $evidence$ is the key contents from the image.

Considering one's internal process when answering questions based on image content, it is typical to decompose the problem into two steps. For example, as in Figure 2, there is a question about "Is the cup near dog in the image". Firstly identify the key elements in the image as evidence ("dog at the pixel position of $\langle 0, 1, 441, 417 \rangle$, cup at the position of $\langle 0, 284, 133, 424 \rangle$, newspaper at the position of $\langle 308, 140, 638, 421 \rangle$, and dog is near the cup, dog is on the table, newspaper is on the table"). Then, combine the relevant content ("dog is near the cup") within the evidence to answer the question. After this process, an answer is generated.

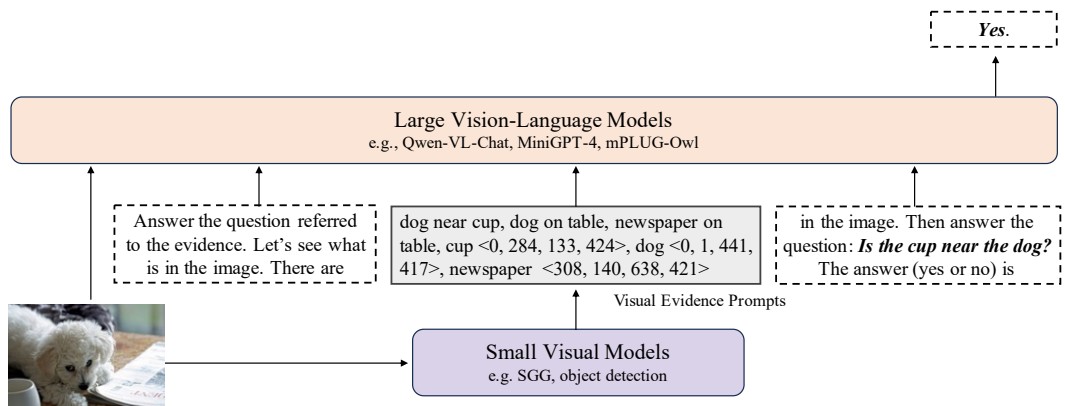

Figure 2: An overview of Visual Evidence Prompting, which mitigates hallucinations in large vision-language models for a text passage via researching visual evidence from small visual models. Given the input image, the small visual models generates visual evidence about different aspects of the image, *e.g.*, object description, object locations and relations between objects. Then the "visual evidence" prompts are used to extract the answer from the image and evidence texts.

## 3.1 TWO-STEP PROMPTING

In this paper, visual models refer to the object detection and scene graph generation models. Other models such as semantic segmentation and human-object interaction can also be considered as visual models and may bring potential performance gains, but that is not the priority of this paper.

**1st step: visual evidence extraction**. The input image of the large vision-language model is fed into the small visual model, and the output is formulated as predefined formats. For the object detection models, the output is composed of the object label and the coordinates of the up-left and down-right point of the bounding box. The visual evidence of object detection is formulated as

$$\{\texttt{label}\}, \langle x_{up\_left}, y_{up\_left}, x_{down\_right}, y_{down\_right}\rangle. \tag{1}$$

For example, the "cup" at $(0, 1, 441, 417)$ is formulated as "cup $\langle 0, 1, 441, 417\rangle$. For the scene graph generation models, the output is composed of the $\langle$subject, relation, object$\rangle$ triplets. Each triplet is firstly formulated as $\{\texttt{subject}\}\{\texttt{relation}\}$ $\{\texttt{object}\}$. Multiple triplets are joined with the ",". For example, $\langle$man on surfboard$\rangle$ and $\langle$man has hair$\rangle$ are formulated as "man on surfboard, man has hair". All the visual evidence are formulated as "there may be evidence in the image too". This is one simple and effective formulation of visual evidence. More sophisticated formats may bring further improvement.

**2nd step: visual evidence prompting.** In the second step, we use symbolized visual evidence along with prompted question to extract the final answer from large vision-language model. To be concrete, we simply concatenate two elements as with "Answer the question referred to the evidence. Let's see what is in the image. There are $\{\texttt{evidence}\}$ in the image. Then answer the question : $\{\texttt{question}\}$? The answer (yes or no) is ". The prompt for this step is self-augmented, since the prompt contains the visual evidence generated by the visual model. Finally, the vision-language model is fed the prompted text as input to generate final answers.

## 3.2 DISCUSSION

As a new paradigm, visual evidence prompting has multiple attractive advantages.

1. First, previous work has found that the frequent occurrence of objects in the instruction datasets are prone to be hallucinated by LVLMs (Li et al., 2023b). The visual evidence is produced by independent visual models, making it immune to the biased statistics.

2. Second, since the evidence is written in natural language formats, it provides an interpretable interface to communicate with large vision-language models (Brown et al., 2020). This paradigm makes it much easier to incorporate visual knowledge into vision-language models by changing visual models and the corresponding visual evidences.

3. Finally, this could not only greatly reduce the computation costs for instruction tuning on new datasets, but also introduces small visual models to the vision-language-model-as-a-service (Sun et al., 2022) and can be easily applied to large-scale real-world tasks.

Moreover, complementing the universal semantic understanding of large vision-language models with the expertise of small visual models, visual evidence prompting has several appealing properties.

1. First, the effectiveness of this method does not heavily rely on the quality of small visual models (Gao et al., 2022). It is expected that small model with larger capacity brings more gains, but a fairly good visual model is enough to bring notable improvement (Sec. 4.3.1).

2. Second, benefiting from the strong semantic understanding of the LVLMs, our method possesses an amount of robustness even provided with erroneous evidence (Chen et al., 2023a) (Sec. 4.3.1).

3. Finally, along with the reasoning-instruction like "referring to the evidence" in the prompt templates, the visual evidence is able to consistently mitigate the hallucination effect (Kojima et al., 2022) (Sec. 4.3.2).

Theses properties are also studied and verified in the following experimantal sections.

# 4 EXPERIMENTS

## 4.1 EXPERIMENTAL SETUP

### 4.1.1 DATASET

In this study, we introduce a novel dataset for the evaluation of relation hallucination. Additionally, we employ datasets proposed by (Li et al., 2023b) to assess object hallucination.

We use the Visual Genome (Krishna et al., 2017) to generate dataset for assessing relation hallucinations. 1) The 50 relation categories of VG are categorized into two groups, spatial and action relationships. 2) We select 7 representative spatial relations and 9 head action relations, while the other tail relations are ignored. 3) For each relation, we randomly select 75 images with questions whose answers are "Yes" and 75 images questions whose the answer are "No". Each "Yes" questions are constructed from annotations. For questions with the answer "No", the probing relations are randomly selected within the corresponding group of spatial or action relations. To ensure not to select synonyms of the ground truth as probing relations, we carefully devise pairs of synonymous relations as the "blacklist". In summary, this dataset consists of 2400 triplets of image, question and answer, in which 1200 are "Yes" and 1200 are "No". There is no overlap between the datasets used for training small models and the LVLM hallucination test sets. We use the POPE's dataset for object hallucination evaluation which use the validation set of COCO. For relation hallucination we follow POPE and use the test set of Visual Genome to construct dataset. More details are shown in the Appendix B.

### 4.1.2 EVALUATION METRIC

Following the evaluation metric in POPE (Li et al., 2023b), we formulates the evaluation of object and relation hallucination as a binary classification task that prompts LVLMs to output "Yes" or "No", e.g., "Is there a chair in this image?" and "Is the cup near the dog in the image?". If the model's response does not include neither "Yes" nor "No", it will be disregarded in the calculation of metrics. We report the accuracy, F1 score and "Yes" ratio. Specifically, the accuracy reflects the proportion of correctly answered questions, while the F1 score combines both results of precision and recall. Besides, we also report the ratio that LVLMs answer "Yes" as the reference to analyze the model behaviors. If the proportion of "Yes" is too high (close to 1) or too low (close to 0), it indicates that the model's predictions are highly biased towards the "Yes" or "No".

### 4.1.3 IMPLEMENTATION DETAILS

In order to conduct our experimental analysis, we select three prominent LVLMs as representatives. These include MiniGPT-4 (Zhu et al., 2023), mPLUG-Owl (Ye et al., 2023) and Qwen-VL-Chat

(Bai et al., 2023). Firstly, we use the corresponding visual small model to process the images in the evaluation dataset and obtain the corresponding evidence. For the evaluation of object hallucination, we can obtain the category and coordinates of objects in the image. For the evaluation of relational hallucination, we can obtain the relationship between objects. Subsequently, the acquired visual evidence is integrated with the original question using a prompt. Note that in Qwen-VL-Chat, for any given bounding box, a normalization process is applied (within the range $[0, 1000)$). For example, $x_{input} = x_{pixel}/width * 1000, y_{input} = y_{pixel}/height * 1000$. So the scale of the image does not differentiate any object relations. Finally, we input the combined prompt and image into the model to obtain the model's answer. We employed the default parameter settings provided in the official repository for Qwen-VL-Chat and mPLUG-Owl models. In the case of MiniGPT-4, we observed that the default temperature coefficient led to an unstable output from the model. To address this issue, we set the temperature value to 0.1 for all of our experiments.

## 4.2 RESULTS

### 4.2.1 EFFECT OF VISUAL EVIDENCE PROMPTING

We evaluate the object and relation hallucination performance of the three baseline models and their counterparts with visual evidence prompting, as well as a state-of-the-art method using instruction tuning named LRV-Instruction (Liu et al., 2023b). The evaluation results are presented in Table 16.

| Evaluation | Model | Accuracy | F1 Score | Yes (%) |
|---|---|---|---|---|
| *Object Hallucination* | mPLUG-Owl | 57.29 | 68.97 | 87.47 |
| | + LRV-Instruction | 68.08 | 72.19 | 64.61 |
| | + Visual Evidence | **78.38** | **78.60** | 50.96 |
| | MiniGPT-4 | 70.89 | 71.01 | 50.38 |
| | + LRV-Instruction | 62.94 | 71.67 | 80.77 |
| | + Visual Evidence | **80.23** | **81.77** | 58.43 |
| | Qwen-VL-Chat | 81.23 | 81.46 | 51.23 |
| | + Visual Evidence | **87.70** | **87.11** | 45.43 |
| *Relation Hallucination* | mPLUG-Owl | 62.58 | 71.18 | 79.83 |
| | + LRV-Instruction | 64.07 | 58.17 | 35.89 |
| | + Visual Evidence | **68.46** | **73.01** | 66.88 |
| | MiniGPT-4 | 64.55 | 72.24 | 77.61 |
| | + LRV-Instruction | 65.41 | 72.71 | 75.88 |
| | + Visual Evidence | **70.11** | **74.01** | 64.99 |
| | Qwen-VL-Chat | 63.62 | 46.99 | 18.62 |
| | + Visual Evidence | **75.68** | **76.69** | 54.40 |

Table 1: Detailed object and relation hallucination evaluation results using POPE evaluation metrics. "+ LRV-instruction" denotes the model from (Liu et al., 2023b). "+ Visual Evidence" denotes ours.

**Object hallucination** The upper part of Table 16 presents the comparison with baseline and SOTA results for the evaluation. After incorporating visual evidence prompting, all models enables more precise discernment of object presence within the image. Notably, the instruction tuning approach (LRV-Instruction (Liu et al., 2023b)) outperforms baseline model on mPLUG-Owl, while underperforms baseline model on MiniGPT-4. This verifies that the instruction tuning method is prone to overfitting to specific datasets, tasks, or models, leading to relatively poor generalization ability, the phenomenon is also studied in (Zhai et al., 2023).

**Relation hallucination** As in the lower part of Table 16, our model is also evaluated on the proposed relation benchmark. Firstly, along with object, our proposed methodology substantiates its efficacy by inducing diverse degrees of enhancement across all models. Secondly, compared with object hallucinations, it is evident that the performance of all models is inadequate in terms of relation. Interestingly, the model Qwen-VL-Chat, which exhibited exceptional performance on the object evaluation (87.7% accuracy and 87.11 F1 score), displayed significant illusion phenomena on the relation dataset, achieving an accuracy of 63% and an F1 score of 46. With the help of visual evidence prompting, the accuracy experienced a notable improvement of 12%, while the F1 score showed a substantial enhancement of nearly 30. These outcomes strongly validate the effectiveness and efficacy of our proposed approach. Thirdly, we observe that the improvement for different models is

directly proportional to their inherent performance. For example, the performance of Qwen-VL-Chat is from 63.62% to 75.68% (+12.07%), while MiniGPT-4 is from 64.55% to 70.11% (+5.56%). We attribute this phenomenon to the original abilities of different models for relations comprehension.

## 4.3 DETAILED ANALYSIS

| Model | Visual model | | Performance of LVLMs | | |
|---|---|---|---|---|---|
| | Model name | mAP | Accuracy | F1 Score | Yes (%) |
| mPLUG-Owl | - | - | 57.29 | 68.97 | 87.47 |
| | yolos-tiny | 28.7 | **70.23** | **70.34** | 50.44 |
| | yolos-small | 36.1 | **73.44** | **73.22** | 49.23 |
| | detr-resnet-50 | 42.0 | **76.55** | **76.72** | 50.57 |
| | detr-resnet-101 | 43.5 | **78.38** | **78.60** | 50.96 |
| Qwen-VL-Chat | - | - | 81.23 | 81.46 | 51.23 |
| | yolos-tiny | 28.7 | **83.73** | **82.25** | 41.67 |
| | yolos-small | 36.1 | **85.47** | **84.35** | 39.23 |
| | detr-resnet-50 | 42.0 | **87.10** | **86.36** | 44.57 |
| | detr-resnet-101 | 43.5 | **87.70** | **87.11** | 45.43 |

Table 2: Object hallucination results of Qwen-VL-Chat and mPLUG-Owl incorporating visual evidence from different object detection models, *i.e.* yolos-tiny (Fang et al., 2021), yolos-small (Fang et al., 2021), detr-resnet-50 (Carion et al., 2020) and detr-resnet-101 (Carion et al., 2020). The mAP on COCO 2017 validation of different visual models is also reported.

### 4.3.1 VISUAL MODEL

**How do LVLMs perform with visual models of different capacities?** In this study, we explore the influence of incorporated visual models with different performance. Specifically, we select Qwen-VL-Chat and mPLUG-owl as the subjects of our investigation. For the object hallucination evaluation, we employ three object detection models with different architectures and capacities, namely yolos-tiny (Fang et al., 2021), yolos-small, detr-resnet-50(Carion et al., 2020), and detr-resnet-101, as our visual models. The experimental results are presented in Table 2. For the relationship hallucination evaluation, we utilize three popular scene graph generation models with different architectures, namely RelTR (Cong et al., 2023), MOTIFS (Zellers et al., 2018) and OpenPSG (Yang et al., 2022). The experimental results are presented in Table 15 in the appendix.

Noteblely, the results of the object hallucination experiments demonstrate a positive correlation between the detection abilities of the chosen visual small models and the performance of object hallucination in LVLMs. As show in the Table 12 and Table 13 in the appendix, both the Qwen-VL-Chat and mPLUG-owl models consistently manifest similar trends across all three object hallucination datasets. This tread is expected as a good detection model provides high-quality visual evidence, which enables better mitigation of objects hallucinations. Interestingly, Qwen-VL-Chat demonstrates comparable performance improvements when employing detr-resnet-101 and detr-resnet-50 as visual models across three object hallucination datasets. The detr-resnet-101 brings more performance gains for mPLUG-Owl, with repect to to detr-resnet-50. The reason for this phenomenon is that performance of Qwen-VL-Chat is saturated with limited room for improvement, while mPLUG-Owl has relatively lower performance but greater chance for enhancement.

**How do LVLMs perform with erroneous visual evidence?** Although the small vision model has attained a commendable level of accuracy within its specific domain, it occasionally exhibits imperfections, such as instances where the object detection model fails to identify inconspicuous objects present in the image, or detects objects that do not actually exist in the image. Consequently, it is neccesary to validate the robustness against the visual evidence.

In this study, we employ the object hallucination results of Qwen-VL-Chat combined with detr-resnet-101 for analysis. Figure 3 present the ratios of samples which are integrated with erroneous visual evidence. It is split as four parts based on the original behavior and the behavior after introducing erroneous evidence. For example, "LVLM wrong answer + wrong evidence → Correct answer" denotes the samples that were initially answered incorrectly and answer correct after provided with

wrong evidence. Firstly, the total ratio of erroneous evidence is $6.9\%$, while the one of correct evidence is $93.1\%$. It illustrates that the effect of erroneous evidence is relatively small in current setting. But this study is still important for the open-set real-world cases. Secondly, after integrating with the incorrect evidence, most of the samples with wrong original answer remain wrong (the first and second columns). This observation is expected as erroneous evidence doesn't provide valuable information. Thirdly, for a substantial fraction of the samples with original correct answers, the model continues to provide correct answers, indicating a certain level of robustness in the model.

Additionally, after in-depth analysis of these cases, we found that whether the model exhibits consistent adherence to the correct answer is related to the prominence of the interested object. If the queried object in the question is relatively prominent in the image, the correctness of the evidence doesn't effect the final answer. Otherwise, the model tends to produced answer conditioned on the evidence. If the evidence is wrong, the response to question is easily to be wrong. Figure 7 and 8 in the appendix showcases some examples.

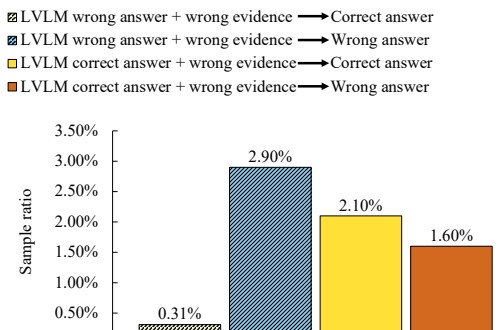

Figure 3: Robustness against correctness of visual evidence with Qwen-VL-Chat and detr-resnet-101. For example, "LVLM wrong answer + wrong evidence → Correct answer" denotes the samples that were initially answered incorrectly and answer correct after provided with wrong evidence.

### 4.3.2 How do LVLMs perform with different prompt templates?

To validate the robustness of visual evidence prompting against input prompts, we evaluate Qwen-VL-Chat with $4$ templates on object and relation evaluations, respectively. Table 3 and 20 summarizes performance. The results indicate that the performance is improved if the text is written in a way that encourages referring the evidence. The difference in accuracy is significant depending on the sentence. In this experiment, the one with more reasoning style achieves the best results. Interestingly, the $5_{th}$ template in Table 3 adds a new prompt which tells the LVLMs that the evidence may be wrong or missing. Compared with the $4_{th}$ template, the improvement of $0.77\%$ accuracy mainly comes from the addressing the "LVLM correct answer + wrong evidence → Wrong answer". In contrast, when we use misleading or irrelevant templates, the performance does not improve.

| Visual Evidence Prompt Templates | Accuracy | F1 Score |
|---|---|---|
| {question} | 81.23 | 81.46 |
| There are {evidence} in the image.\n{question} | 87.70 | 87.11 |
| {evidence} are existing in the image.\n{question} | 86.37 | 85.70 |
| You can see {evidence} in the image.\n{question} | 85.80 | 85.30 |
| The following object are existing in the image: {evidence}.\n{question} | **87.10** | **87.11** |
| The following object are existing in the image: {evidence}. The evidence might be wrong. Keep your answer if you think the evidence is wrong or evidence is missing. \n{question} | **87.87** | **87.31** |
| It's a beautiful day.\n{question} | 77.23 | 72.97 |
| There is nothing in the image.\n{question} | 68.37 | 55.51 |

Table 3: Robustness study against template measured on the object dataset on Qwen-VL-Chat.

### 4.3.3 Fine-grained results of relation hallucination

In this section, we comprehensively demonstrate and analyze the model's performance across diverse relationship categories. In Figure 4, the performance of Qwen-VL-Chat with and without corresponding visual evidence is presented for each relationship category in the relation hallucination dataset, where spatial relationships are depicted on the left and action relationships on the right. Based on the

depicted results, it is evident that Qwen-VL-Chat exhibits varying degrees of improvement across different relationship categories following the integration of visual evidence. Notably, a significant enhancement is observed in the action relationship category. Overall, the model outperforms the spatial relationship in the context of action relationships. This discrepancy could be attributed to the finer-grained nature of spatial relationships within images, which demand a higher level of comprehension capability.

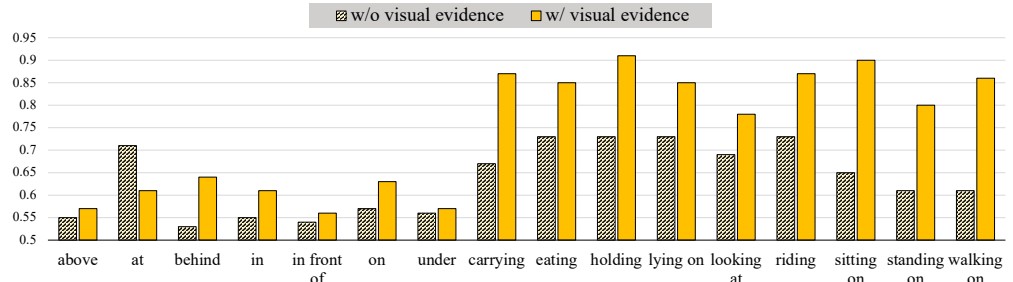

Figure 4: The effect of incorporating visual evidence on the performance of Qwen-VL-Chat across different relation categories in the proposed relation hallucination dataset has been presented in this figure.

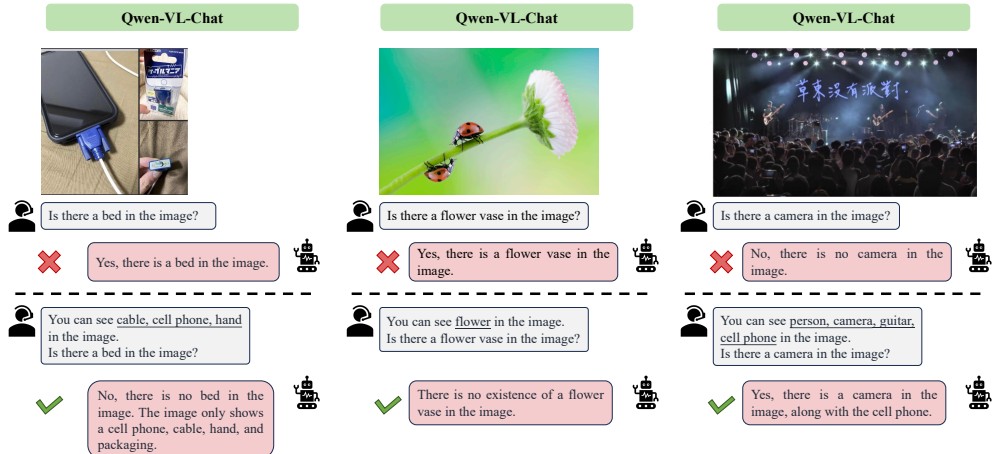

Figure 5: This figure showcases several instances of object hallucination mitigation of Qwen-VL-Chat facilitated by our framework. The dialogue above the dashed line depicts scenarios without visual evidence, whereas the dialogue below the dashed line includes visual evidence. The included images are sourced from (OpenAI, 2023), winoground (Thrush et al., 2022). and MiniGPT-4 (Zhu et al., 2023) respectively

### 4.3.4 How do visual evidence performs on out-of-domain images?

As shown in Figure 5, the combination of visual evidence can reduces both object and relation hallucination when testing on out-of-domain image, which demonstrate the robustness and generalization ability of our method. More cases are shown in the Figure 13 in the appendix.

## 5 Conclusion

We have explored visual evidence prompting as a simple and broadly applicable method for mitigating hallucinations in large vision-language models. Through experiments on object and the proposed relation benchmark, we find that visual evidence is a effective, robust and general cure for large vision-language models. Broadening the range of visual evidence that will hopefully inspire further work on external-knowledge-based approaches for mitigating hallucinations.

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

# A    APPENDIX: DISCUSSIONS

## LIST OF REVISIONS

## A.1    OUT-OF-DOMAIN DATA

Reviewer rr7K, Q2

Reviewer w1s8, Q1

We also conduct more evaluations on more out-of-domain datasets. Specifcally, we collect 2,540 additional samples from open-world datasets and scenarios to further evaluate the genaliation ability of our method. We collect 2,540 samples from another two object detection (Object365 (Shao et al., 2019)) and scene graph generation (OpenImage (Kuznetsova et al., 2020)) datasets for quantitative analysis. The Object365 dataset is a collection of images that aims to provide a comprehensive representation of objects commonly found in indoor environments. It consists of over 365 object categories, with each category containing multiple images depicting different instances of the object. The OpenImage dataset consists of millions of images covering a wide range of categories, including objects, scenes, and events. It provides valuable annotations for each image, including object bounding boxes, class labels, and object relationships. Table 4 present the comparison with baseline results for the evaluation on out-of-domain datasets. The experimental results indicate that in out-of-domain open scenarios, incorporating visual evidence can still mitigate the hallucination of LVLMs significantly.

In $20.6\%$ of the images, small model captures incorrect or partial correct object or relation information. With these visual evidences, only $8\%$ of the false evidence confuse the LVLM and change the response from collect to wrong. Finally, we would like to point out that the contribution of our method lies in combining small and large models, utilizing the domain-specific knowledge of small models to complement the large models. In practical applications, it is possible to customize domain-specific small models to tailor different domain knowledge.

## A.2    WHY NOT FINETUNING?

Reviewer w1s8, Q5

It is a common practice to fine-tune foundation models on specific tasks to enhance task performance or align the model's behavior with human expectations. It is well-known that the foundation models gain speciality to achieve exceptional performance on the fine-tuning task, but it can potentially lose its generality. This phenomenon is closely associated with the concept of catastrophic forgetting observed in deep neural networks.

| Evaluation | Model | Accuracy | F1 Score | Yes (%) |
|---|---|---|---|---|
| *Object Hallucination (Out-of-domain)* | mPLUG-Owl | 52.04 | 66.81 | 94.52 |
| | + Visual Evidence | **62.46** | **69.43** | 72.67 |
| | Qwen-VL-Chat | 70.25 | 60.31 | 24.95 |
| | + Visual Evidence | **76.74** | **75.50** | 45.01 |
| *Relation Hallucination (Out-of-domain)* | mPLUG-Owl | 58.52 | 69.06 | 84.07 |
| | + Visual Evidence | **72.41** | **75.29** | 66.88 |
| | Qwen-VL-Chat | 73.93 | 71.54 | 41.71 |
| | + Visual Evidence | **75.98** | **72.84** | 38.18 |

Table 4: Detailed object and relation hallucination evaluation results on out-of-domain datasets constructed from Object365 (2000 samples) and OpenImage (540 samples).

Previous work Zhai et al. (2023) has conducted fine-tuning experiments on LLaVA. As the fine-tuning progresses, LLaVA starts to hallucinate by disregarding the questions and exclusively generating text based on the examples in the fine-tuning datasets. As in the Table 3 in Zhai et al. (2023), after 1 epoch finetuning LLaVA-7b on MNIST, the accuracy on CIFAR-10 significantly drops from $56.71\%$ to $9.27\%$. On the other hand, our prompt-based method does not modify the parameters of the model, and offer greater controllability, which is advantageous for preserving the model's original generalization capability.

### A.3   OVERLAP BETWEEN OBJECTS IN EVIDENCE AND QUESTIONS

Reviewer YNjN, Q1

| | correct $\rightarrow$ correct | correct $\rightarrow$ wrong | wrong $\rightarrow$ correct | wrong $\rightarrow$ wrong |
|---|---|---|---|---|
| Type A | 139 (46.7%) | 8 (2.7%) | 110 (36.9%) | 41 (13.8%) |
| Type B | 415 (34.5%) | 22 (1.8%) | 563 (46.8%) | 202 (16.8%) |

Table 5: Robustness against the overlap between objects in questions and objects in evaluation datasets' questions. For example, "wrong $\rightarrow$ correct" denotes the samples that were initially answered incorrectly and answer correct after provided with visual evidence.

We calculate the current stats of the overlap between objects in questions and objects in evaluation datasets' questions. In the 3,000 visual evidence prompts, there are 298 prompts that contains object that are not in the question (Type A), and 1,202 prompts that contain objects that are not in the questions exclusively (Type B).

Following Figure 3, we calculate the stats of samples which were initally answered corrently/wrongly and answer correctly/wrongly after provided with Type A/B prompts (Table 5. In the 298 Type A prompts, 110 of which ($36.9\%$) allievates the hallucination of LVLM with detr-resnet-101 on Qwen-VL-Chat. In the $1,202$ Type B prompts, 563 of which ($46.8\%$) allievates the hallucination of LVLM.

| Model | Setting | Accuracy |
|---|---|---|
| mPLUG-Owl | baseline | 57.29% |
| | + visual evidence | 78.38% |
| | + visual evidence to synonyms | 71.54% |
| Qwen-VL-Chat | baseline | 81.23% |
| | + visual evidence | 87.70% |
| | + visual evidence to synonyms | 86.53% |

Table 6: Robustness against the object labels to synonyms.

| Model | baseline | + object labels | + relation labels |
|---|---|---|---|
| mPLUG-Owl | 63.62% | 71.41% | 75.68% |
| Qwen-VL-Chat | 62.58% | 66.88% | 68.46% |

Table 7: Only use object labels as visual evidence for relation hallucination

| Model | Setting | In-domain objects | Out-of-domain objects |
|---|---|---|---|
| mPLUG-Owl | Baseline | 58.68% | 48.45% |
| | + Visual Evidence | 65.38% | 60.87% |
| Qwen-VL-Chat | Baseline | 74.64% | 67.87% |
| | + Visual Evidence | 79.77% | 75.10% |

Table 8: Performance on open-vocabulary objects.

With the help of ChatGPT, we also manually change the object appear in question to its synonyms respectively. The evaluation of object hallucination slightly decreases from 87.70% to 86.53% on Qwen-VL-Chat and from 78.38% to 71.54% on mPLUG-Owl, but there is still a non-trival improvement over the baseline especially on mPLUG-Owl, the results are shown in the Table 6.

### A.4 ONLY OBJECT LABELS

Reviewer bJ8t, Q1

We conduct validation experiments using the detr-resnet-101 model to provide object labels as evidence for relation hallucination.

The results in Table 7 show that providing object labels as evidence also has some improvement although not as effective as relation label. We suppose it is because object labels themselves contain crucial object information from the image, which leads to mitigating relation hallucination. This result not only validates the necessity of relation labels but also further verifies that our approach is orthogonal to the specific task.

### A.5 OPEN-VOCABULARY OBJECTS AND FEW-SHOT RELATIONS

Reviewer bJ8t, Q2

We construct a new out-of-domain object hallucination dataset with 2000 samples using the test sets from Object365 (Shao et al., 2019) following the construction idea of POPE Li et al. (2023b). This dataset is divided into two parts. One part includes 80 objects that defined in COCO, while the other portion consists of objects that do not appear in COCO. The performance of these two parts are shown in the Table 8. It can be observed that there is a consistent improvement in performance for both in-domain and out-of-domain object categories.

We chose the bottom-10 tail relations as defined in (He et al., 2020) of VG to construct a medium-sized relation hallucination dataset with 1006 samples. We used OpenPSG as the SGG model and conducted experiments on Qwen-VL-Chat and mPLUG-Owl. The experiment results are shown in the table 9 below, and it can be seen that our framework still achieves significant improvements in few-shot relations.

## B    APPENDIX: DATASETS

**Object hallucination datasets.** In POPE(Li et al., 2023b), 500 images are randomly selected from the validation set of COCO Vinyals et al. (2016), with more than three ground-truth objects in the annotations. For each image, 6 questions are constructed from annotations whose answers are "Yes". For questions with the answer "No", three strategies, *i.e.*, Random, Popular, Adversarial, are considered to sample their probing objects. The difficult of question increases from Random to

| Model | Setting | Accuracy (%) |
|---|---|---|
| mPLUG-Owl | Baseline | 61.23% |
| | + Visual Evidence | 67.89% |
| Qwen-VL-Chat | Baseline | 55.67% |
| | + Visual Evidence | 68.63% |

Table 9: Performance on few-shot relations.

Adversarial. For MSCOCO-Random, objects that do not exist in the image are randomly sampled. For MSCOCO-Popular, the top-3% most frequent objects in MSCOCO are selected. For MSCOCO-Adversarial, first rank all the objects according to their co-occurring frequencies with the ground-truth objects, and then select the top-k most frequent ones that do not exist in the image.

**Relation hallucination datasets.** Firstly we categorize the all 50 relationships into two groups, *i.e.*spatial and action relationships. The spatial relation categories include above, at, behind, in, in front of, near, on, on back of, over, underand laying on. The spatial relationship categories consist of carryingcovered incoveringeatingflying ingrowing onhanging from holdinglying onlooking atmounted onparked on, ridingsitting onstanding on, walking onwalking in, and watching. Subsequently, we proceed to select 7 spatial relationships, specifically above, at, behind, in, in front of, on, and under, as well as 9 action relationships, namely carrying, eating, holding, lying on, looking at, riding, sitting on, standing on, and walking on. For each relation, we randomly select 75 images with questions whose answers are "Yes" and 75 images questions whose the answer are "No". Each "Yes" questions are constructed from annotations. For questions with the answer "No", the probing relations are randomly selected within the corresponding group of spatial or action relations with additional added negative relation, which is shown in the Table 10. To ensure not select synonyms of the ground truth as probing relations, we carefully devise several pairs of synonymous relations as the "blacklist" as shown in the Table 10. Finally the dataset consists of 2400 triplets of image, question and answer, in which 1200 are "Yes" and 1200 are "No". In Figure 6, we show some cases in our dataset.

| Relation type | Negative relations | Synonymous pairs |
|---|---|---|
| *Spatial relation* | **above, at, behind, in, in front of, on, under**
*at the left of, at the right of* | above: {on}
on: {above} |
| *Action relation* | **carrying, eating, holding, lying on, looking at, riding, sitting on, standing on, walking on**
*walking in, watching, cutting, feeding, leaning on, jumping over, hugging, kissing, pushing, pulling, washing, kicking, draging* | walking on: {walking in, standing on}
looking at: {watching} |

Table 10: The negative relations candidate set used to contruct negative question are shown here. We also present the synonymous pairs used to ensure not select synonyms of the ground truth as probing relations

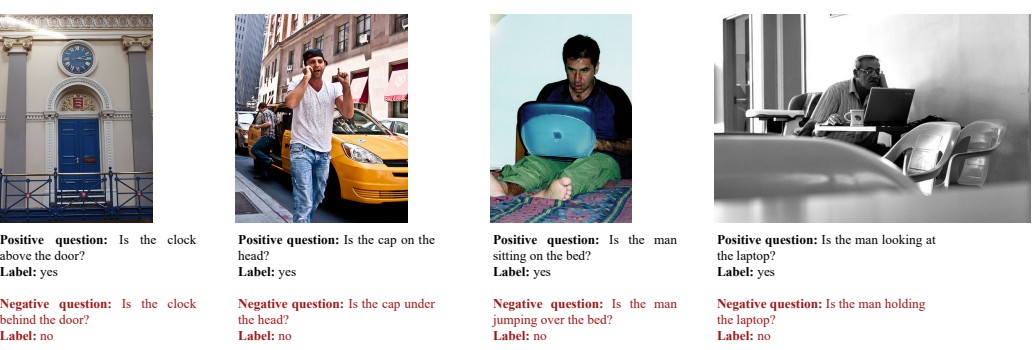

**Positive question:** Is the clock above the door?
**Label:** yes

**Negative question:** Is the clock behind the door?
**Label:** no

**Positive question:** Is the cap on the head?
**Label:** yes

**Negative question:** Is the cap under the head?
**Label:** no

**Positive question:** Is the man sitting on the bed?
**Label:** yes

**Negative question:** Is the man jumping over the bed?
**Label:** no

**Positive question:** Is the man looking at the laptop?
**Label:** yes

**Negative question:** Is the man holding the laptop?
**Label:** no

Figure 6: Several cases in our proposed relation hallucination dataset are depicted in this figure, with the two on the left representing spatial relations and the two on the right illustrating action relations.

## C  APPENDIX: DETAILED OBJECT HALLUCINATION EVLUATION

### C.1  MORE RESULTS ABOUT OBJECT HALLUCINATION

The evaluation results on MSCOCO-Popular and MSCOCO-Random of Qwen-VL-Chat, mPLUG-Owl, MiniGPT-4 and LRV-instruction are presented in Table 11.

| Object hallucination datasets | Model | Accuracy | F1 Score | Yes (%) |
|---|---|---|---|---|
| *MSCOCO-Popular* | mPLUG-Owl | 57.96 | 68.96 | 85.39 |
| | + LRV-Instruction | 71.81 | 74.48 | 60.26 |
| | + Visual Evidence | **77.63** | **77.86** | 50.89 |
| | MiniGPT-4 | 73.67 | 73.04 | 47.67 |
| | + LRV-Instruction | 66.24 | 73.57 | 77.64 |
| | + Visual Evidence | **80.77** | **82.08** | 57.30 |
| | Qwen-VL-Chat | 85.53 | 85.08 | 47.00 |
| | + Visual Evidence | **89.60** | **88.87** | 43.40 |
| *MSCOCO-Random* | mPLUG-Owl | 62.86 | 72.11 | 81.55 |
| | + LRV-Instruction | 79.41 | 80.90 | 55.64 |
| | + Visual Evidence | **81.18** | **81.22** | 48.70 |
| | MiniGPT-4 | 80.85 | 79.33 | 41.08 |
| | + LRV-Instruction | 69.48 | 76.08 | 75.77 |
| | + Visual Evidence | **89.69** | **89.87** | 50.24 |
| | Qwen-VL-Chat | 88.17 | 87.79 | 45.29 |
| | + Visual Evidence | **90.79** | **90.31** | 43.51 |

Table 11: Detailed object hallucination evaluation results of LVLMs on MSCOCO-Popular and MSCOCO-Random using POPE evaluation pipeline.

## D  APPENDIX: DIFFERENT VISUAL MODELS AND LVLMS

### D.1  MORE RESULTS ABOUT THE PERFORMANCE OF LVLMS INCORPORATED WITH VISUAL MODELS OF DIFFERENT CAPACITIES

In Figure 12, Figure 13 and Figure 14, we show more results on the MSCOCO-Popular and MSCOCO-Random about the performance of three LVLMs incorporated with visual models of different capacities. In Figure 15, we present the results on VG Relation Hallucination dataset of Qwen-VL-Chat incorporated with different scene graph generation models. This results demonstrate that different scene graph generation models (RelTR, MOTIFS and OpenPSG) have comparable improvements on mPLUG-Owl and Qwen-VL-Chat. For example, RelTR achieves 5.92% and MOTIFS achieves 5.8% improvement on mPLUG-Owl. RelTR achieves 11.35% and MOTIFS achieves 12.55% improvement on Qwen-VL-Chat. The gains brought by different scene graph generation models to LVLM are within a stable range (saturated).

Reviewer bJ8t, Q4

### D.2  MORE RESULTS ON DIFFERENT LVLMS AND LARGER DETECTION MODELS

Reviewer w1s8, Q4

We have also conducted experiments on LLaVA and LLaVA-1.5 to further validate the effectiveness of our method.

It is observed that the hallucination evaluation of LLaVA-1.5 is indeed state-of-the-art, with an accuracy of 84.47% for object hallucination. However, it still exhibits a significant amount of relation hallucination, with an accuracy of 70.38%. Besides LLaVA, visual evidence prompting further helps LLaVA-1.5 alleviate both object and relation hallucination capabilities $84.47\% \to 90.20\%$, $70.38\%$

| Datasets | Visual model | | Qwen-VL-Chat | | |
| | Model name | mAP | Accuracy | F1 Score | Yes (%) |
| --- | --- | --- | --- | --- | --- |
| *MSCOCO-Popular* | - | - | 85.53 | 85.08 | 47.00 |
| | yolos-tiny | 28.7 | **85.90** | **84.27** | 39.63 |
| | yolos-small | 36.1 | **87.37** | **86.12** | 41.03 |
| | detr-resnet-50 | 42.0 | **89.10** | **88.23** | 42.63 |
| | detr-resnet-101 | 43.5 | **89.60** | **88.87** | 43.40 |
| *MSCOCO-Random* | - | - | 88.17 | 87.79 | 45.29 |
| | yolos-tiny | 28.7 | **86.74** | **85.43** | 39.52 |
| | yolos-small | 36.1 | **88.18** | **87.22** | 40.96 |
| | detr-resnet-50 | 42.0 | **90.10** | **89.49** | 42.61 |
| | detr-resnet-101 | 43.5 | **90.79** | **90.31** | 43.51 |

Table 12: Object hallucination results of Qwen-VL-Chat incorporating visual evidence from different object detection models, *i.e.* yolos-tiny Fang et al. (2021), yolos-small Fang et al. (2021), detr-resnet-50 Carion et al. (2020) and detr-resnet-101 Carion et al. (2020). The mAP on COCO 2017 validation of different visual models is also reported.

| Datasets | Visual model | | mPLUG-Owl | | |
| | Model name | mAP | Accuracy | F1 Score | Yes (%) |
| --- | --- | --- | --- | --- | --- |
| *MSCOCO-Popular* | - | - | 57.96 | 68.96 | 85.39 |
| | yolos-tiny | 28.7 | **70.15** | **70.74** | 51.87 |
| | yolos-small | 36.1 | **73.92** | **73.74** | 49.33 |
| | detr-resnet-50 | 42.0 | **76.24** | **76.67** | 51.62 |
| | detr-resnet-101 | 43.5 | **77.63** | **77.86** | 50.89 |
| *MSCOCO-Random* | - | - | 62.86 | 72.11 | 81.55 |
| | yolos-tiny | 28.7 | **73.06** | **73.03** | 48.38 |
| | yolos-small | 36.1 | **76.61** | **76.73** | 49.00 |
| | detr-resnet-50 | 42.0 | **80.65** | **80.78** | 49.01 |
| | detr-resnet-101 | 43.5 | **81.18** | **81.22** | 48.70 |

Table 13: Object hallucination results of mPLUG-Owl incorporating visual evidence from different object detection models, *i.e.* yolos-tiny Fang et al. (2021), yolos-small Fang et al. (2021), detr-resnet-50 Carion et al. (2020) and detr-resnet-101 Carion et al. (2020). The mAP on COCO 2017 validation of different visual models is also reported.

| Datasets | Visual model | | MiniGPT-4 | | |
| | Model name | mAP | Accuracy | F1 Score | Yes (%) |
| --- | --- | --- | --- | --- | --- |
| *MSCOCO-Popular* | - | - | 73.67 | 73.04 | 47.67 |
| | detr-resnet-50 | 42.0 | **80.70** | **81.99** | 57.17 |
| | detr-resnet-101 | 43.5 | **80.77** | **82.08** | 50.89 |
| *MSCOCO-Random* | - | - | 80.85 | 79.33 | 41.08 |
| | detr-resnet-50 | 42.0 | **89.55** | **88.20** | 49.83 |
| | detr-resnet-101 | 43.5 | **89.69** | **89.87** | 50.24 |

Table 14: Object hallucination results of MiniGPT-4 incorporating visual evidence from different object detection models, detr-resnet-50 Carion et al. (2020) and detr-resnet-101 Carion et al. (2020). The mAP on COCO 2017 validation of different visual models is also reported.

| Model | Visual model | | Performance of LVLMs | | |
|---|---|---|---|---|---|
| | Model name | mAP | Accuracy | F1 Score | Yes (%) |
| mPLUG-Owl | - | - | 62.58 | 71.18 | 79.83 |
| | RelTR | 18.9 | **68.50** | **73.06** | 66.92 |
| | MOTIFS | 20.0 | **68.38** | **73.17** | 67.88 |
| | OpenPSG | 28.4 | **68.25** | **72.82** | 66.83 |
| Qwen-VL-Chat | - | - | 63.62 | 46.99 | 18.62 |
| | RelTR | 18.9 | **74.97** | **75.62** | 52.70 |
| | MOTIFS | 20.0 | **75.80** | **76.44** | 52.77 |
| | OpenPSG | 28.4 | **76.17** | **77.00** | 53.71 |

Table 15: Relation hallucination results of Qwen-VL-Chat and mPLUG-Owl incorporating visual evidence from different scene graph generation models, *i.e.* RelTR (Cong et al., 2023), MOTIFS (Zellers et al., 2018) and OpenPSG (Yang et al., 2022). The Recall@20 on PSG benchmark of different visual models is also reported.

$\rightarrow 75.08\%$, thereby providing further evidence of the effectiveness of our method. This indicates that not only can different small models help alleviate hallucinations in large models, but a single small model can consistently alleviate hallucinations in large models of different sizes and trained on different datasets. This result further confirms the complementarity between large and small models and the necessity of our framework.

We also conduct experiments with larger open-source detection models, DINO (Zhang et al., 2022), which is the top-tier model with 58.0 mAP in COCO 2017 val (detr-resnet-101 has 43.5 mAP). The results are shown in Table 17, it can be observed that as the mAP increases, the small model consistently provides a boost to the large model, though it gradually saturates.

Reviewer w1s8, Q3

| Evaluation | Model | Accuracy |
|---|---|---|
| *Object Hallucination* | LLaVA | 60.23 |
| | + Visual Evidence | **77.43** |
| | LLaVA-1.5 | 84.47 |
| | + Visual Evidence | **90.20** |
| *Relation Hallucination* | LLaVA | 64.49 |
| | + Visual Evidence | **70.54** |
| | LLaVA-1.5 | 70.38 |
| | + Visual Evidence | **75.08** |

Table 16: Object and relation hallucination evaluation results on LLaVA and LLaVA-1.5.

## E   APPENDIX: MORE ABLATIONS

### E.1   VISUAL RESULTS OF ERRONEOUS EVIDENCE

We show some cases where the model insists on the correct answer when wrong visual evidence is provided and some cases where the model was misled by the wrong evidence.

### E.2   MORE RESULTS ABOUT THE ABLATION OF QUESTION TEMPLATES IN OBJECT AND RELATION HALLUCINATION EVALUATION

To verify the stability of our method against different question prompt templates,, As shown in Table 18 and Table 19, under different question templates, Qwen-VL-Chat shows consistent performance gain with low standard deviations in both object hallucination and relation hallucination datasets. Such results further validate the robustness of our method. We also present the results of robustness study of visual evidence prompting against different input prompts on relation evaluation.

| Model | Visual model | | Performance of LVLMs |
| | Model name | mAP | Accuracy |
| --- | --- | --- | --- |
| mPLUG-Owl | - | - | 57.29 |
| | yolos-tiny | 28.7 | 70.23 (+12.94) |
| | yolos-small | 36.1 | 73.44 (+3.21) |
| | detr-resnet-50 | 42.0 | 76.55 (+3.11) |
| | detr-resnet-101 | 43.5 | 78.38 (+1.83) |
| | DINO-4scale-swin | 58.0 | **79.44 (+1.06)** |
| Qwen-VL-Chat | - | - | 81.23 |
| | yolos-tiny | 28.7 | **83.73** |
| | yolos-small | 36.1 | **85.47** |
| | detr-resnet-50 | 42.0 | **87.10** |
| | detr-resnet-101 | 43.5 | **87.70** |
| | DINO-4scale-swin | 58.0 | **89.17 (+1.46)** |

Table 17: Object hallucination results of Qwen-VL-Chat and mPLUG-Owl incorporating visual evidence from larger version object detection model (Liu et al., 2022). The values in parentheses indicate the performance improvement compared to the previous row's setting in the table.

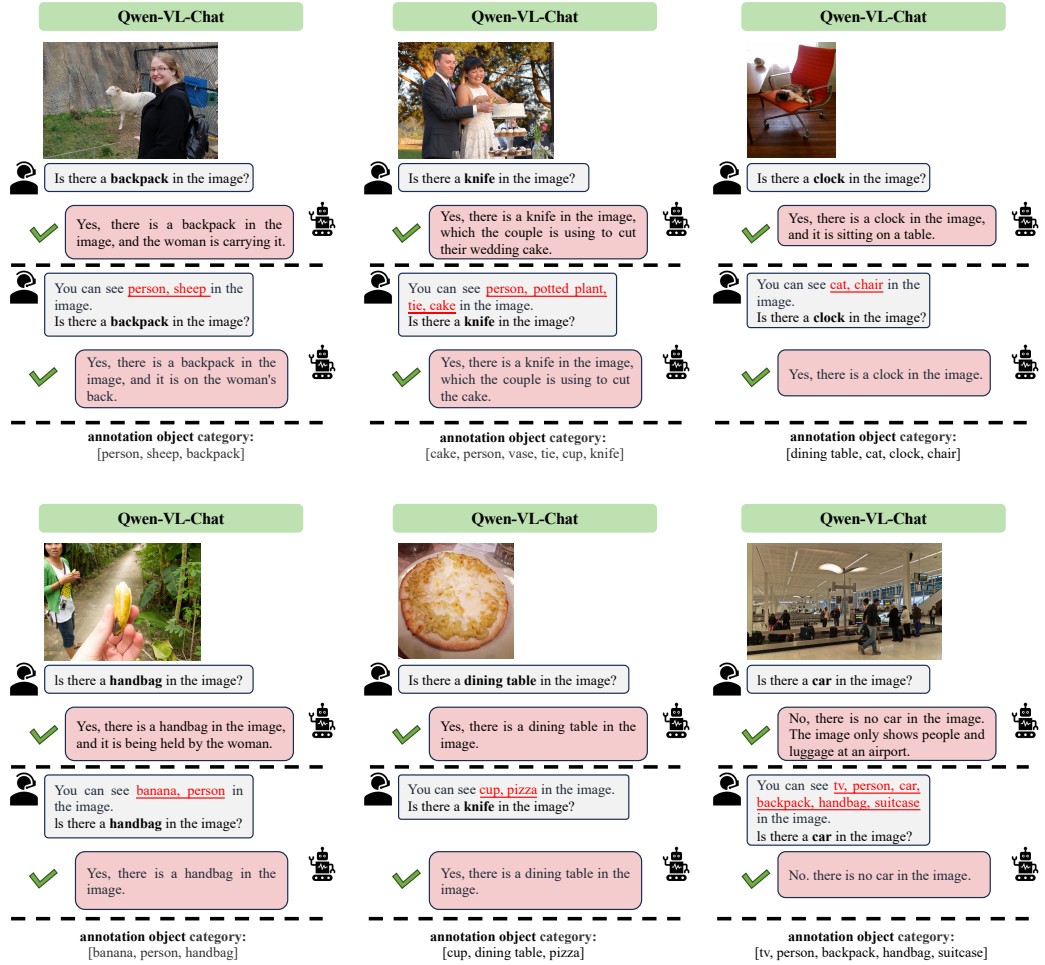

Figure 7: Some cases wherein the initial response was correct, and the model persistently produced correct response despite provided with erroneous visual evidence. The incorrect evidence is underlined in red font. The annotated object categories is presented below each case.

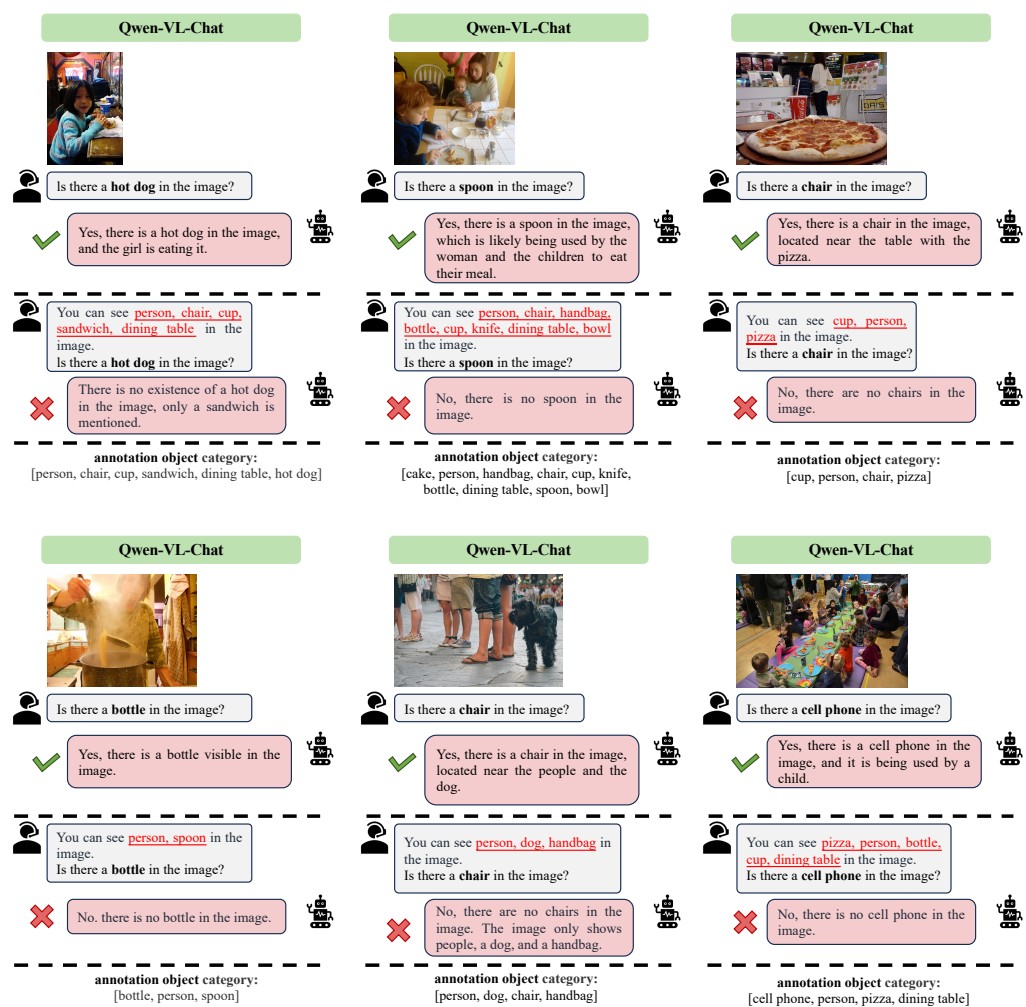

Figure 8: Some cases wherein the initial response was correct, and the model produce wrong response when provided with erroneous visual evidence. The incorrect evidence is underlined in red font. The annotated object categories is presented below each case.

| Question Prompt Templates | Accuracy | | F1 Score | |
| --- | --- | --- | --- | --- |
| | Baseline | Baseline + VE | Baseline | Baseline + VE |
| Is there a <object> in the image? | 80.93 | **87.73** | 81.10 | **86.63** |
| Does the image contains a <object>? | 83.32 | **87.37** | 82.46 | **86.01** |
| Is there any <object> present in the image? | 80.53 | **87.03** | 80.80 | **85.76** |
| Can you see a <object> in the image? | 80.85 | **87.17** | 80.46 | **85.81** |
| Avg±Std. | 81.41±1.11 | **87.32**±0.26 | 81.20±0.76 | **86.05**±0.35 |

Table 18: The evaluation results of Qwen-VL-Chat on MSCOCO-Adversarial before and after incorporating visual evidence across diverse question prompt templates are presented in this table.

| Question Prompt Templates | Accuracy | | F1 Score | |
|---|---|---|---|---|
| | Baseline | Baseline + VE | Baseline | Baseline + VE |
| Is the <subject> <relation> the <object>? | 63.62 | **74.97** | 46.99 | **75.66** |
| Can you see the <subject> <relation> the <object>? | 55.58 | **73.14** | 21.62 | **73.35** |
| Is the <subject> <relation> the <object> in the image? | 64.96 | **74.99** | 51.36 | **74.86** |
| Can you see the <subject> <relation> the <object> in the image? | 57.33 | **73.15** | 27.99 | **72.95** |
| Avg±Std. | 60.37±3.99 | **74.06±0.92** | 36.99±12.48 | **74.21±1.10** |

Table 19: The evaluation results of Qwen-VL-Chat on VG Relation Hallucination dataset before and after incorporating visual evidence across diverse question prompt templates are presented in this table.

| Visual Evidence Prompt Templates | Accuracy | F1 Score |
|---|---|---|
| {question} | 63.62 | 46.99 |
| Evidence: There are {evidence} in the image.\n Let's refer to the evidence and then answer the following question.\n{question} | 74.48 | 75.34 |
| Evidence: You can see {evidence} in the image.\n Let's consider the evidence and then answer the following question.\n{question} | 74.97 | 75.66 |
| Evidence: You can see {evidence} in the image.\n {question} According to the image and evidence, the answer is | 75.27 | 73.48 |
| You can see {evidence} in the image.\n Then answer the question based on what you see: {question} | **75.50** | **78.52** |
| It's a beautiful day.\n{question} | 53.33 | 13.85 |
| There is nothing in the image.\n{question} | 55.56 | 20.68 |

Table 20: Robustness study of Qwen-VL-Chat against template measured on the VG Relation Hallucination dataset.

# F  APPENDIX: CASE STUDY

## F.1  MORE CASES ON OUT-OF-DOMAIN IMAGES

More out-of-domain cases are shown here. Following the idea of CLIP (Radford et al., 2021), we selected 10 out-of-domain datasets from the 27 datasets used to test the zero-shot generalization performance of CLIP. These 10 datasets are Caltech-101 (Fei-Fei et al., 2004), OxfordPets (Parkhi et al., 2012), Birdsnap (Berg et al., 2014), Flowers102 (Nilsback & Zisserman, 2008), CLEVRCounts (Johnson et al., 2017), Country211 (Radford et al., 2021), Food101 (Bossard et al., 2014), SUN397 Xiao et al. (2010), HatefulMemes (Kiela et al., 2020), and STL10 (Coates et al., 2011). Then, we randomly selected two images from each dataset, one for evaluating object hallucination and the other for evaluating relation hallucination. As shown in Figure 11 and Figure 12, we can see that even when providing incorrect visual evidence to the model, it still maintains its original correct answer, which further verifies the model's robustness to incorrect evidence in open scenarios.

Reviewer rr7K, Q2
Reviewer w1s8, Q1

## F.2  SOME CASES ON OBJECT COUNTING AND OCR

In Figure 9 and Figure 10, we also show some cases that other tasks like object counting and OCR can also be enhanced by our framework.

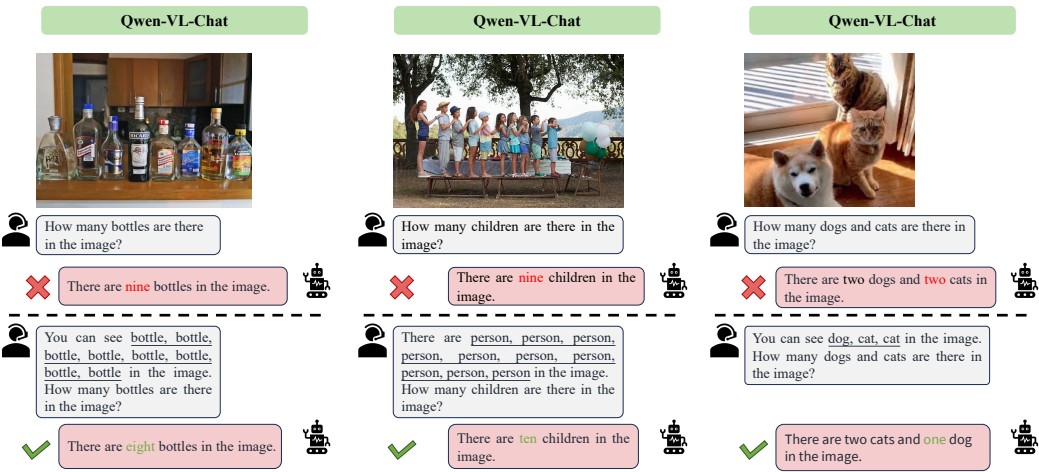

Figure 9: Some cases of Qwen-VL-Chat on object counting task.

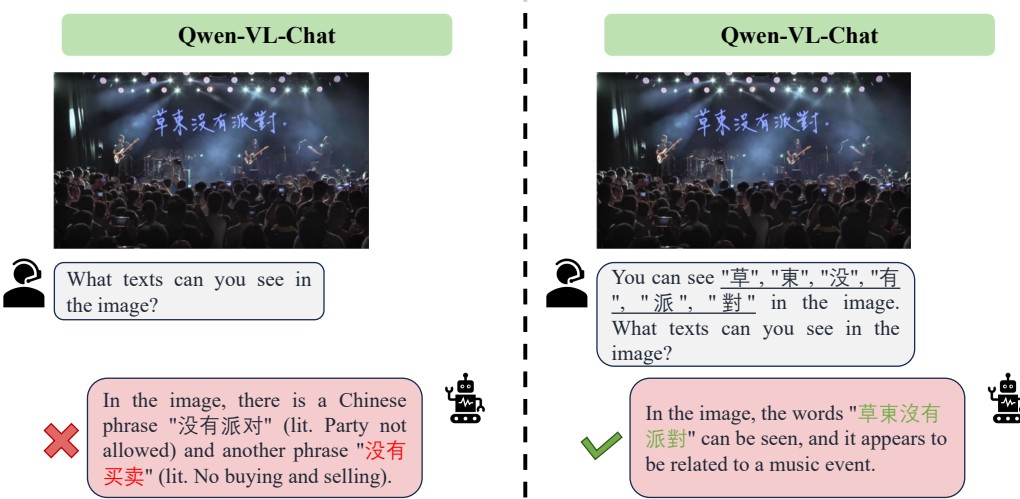

Figure 10: A case of Qwen-VL-Chat on OCR task.

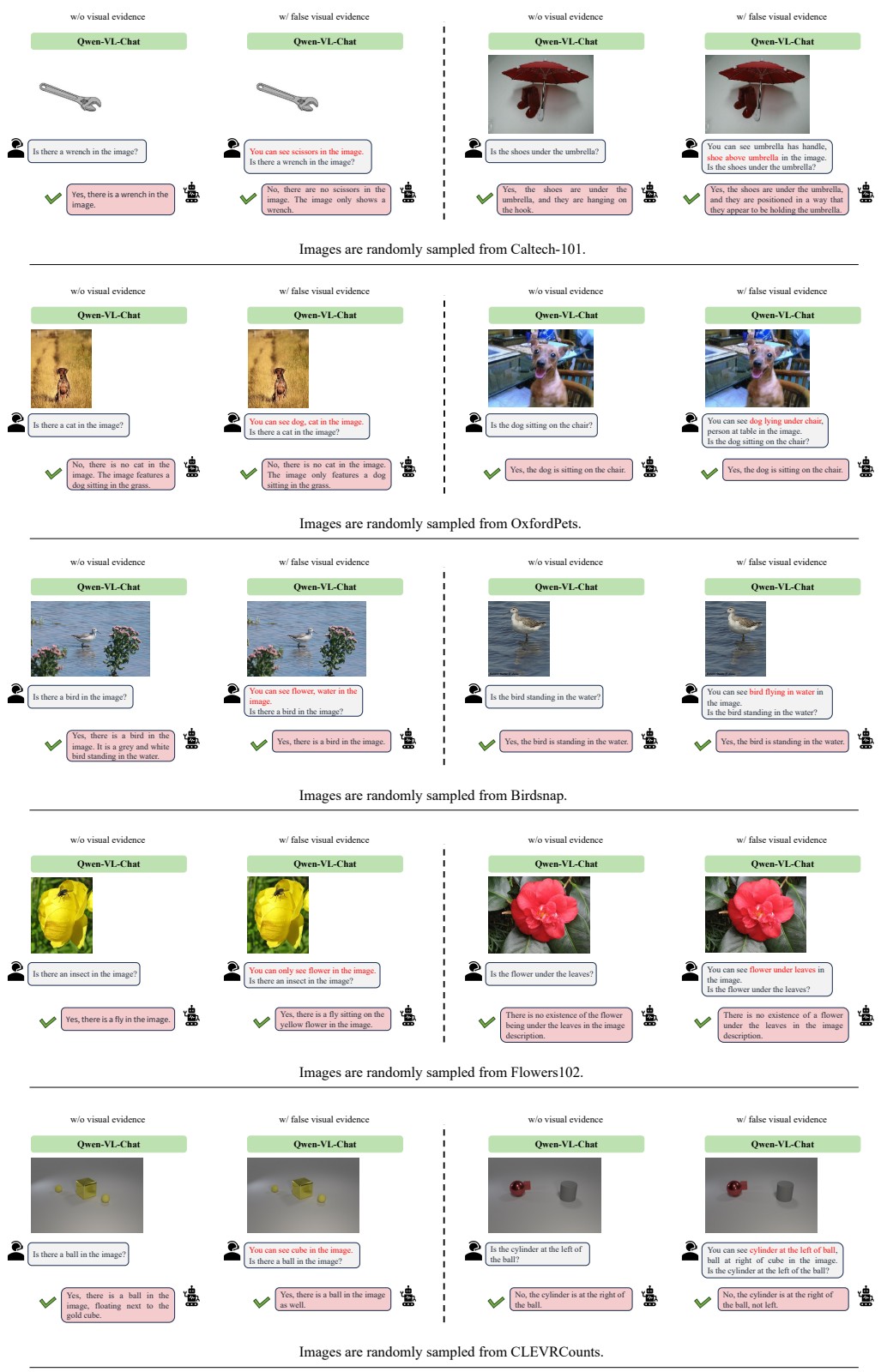

Figure 11: Some open-scenario cases from different out-of-domain datasets when LVLM are provided with false visual evidence.

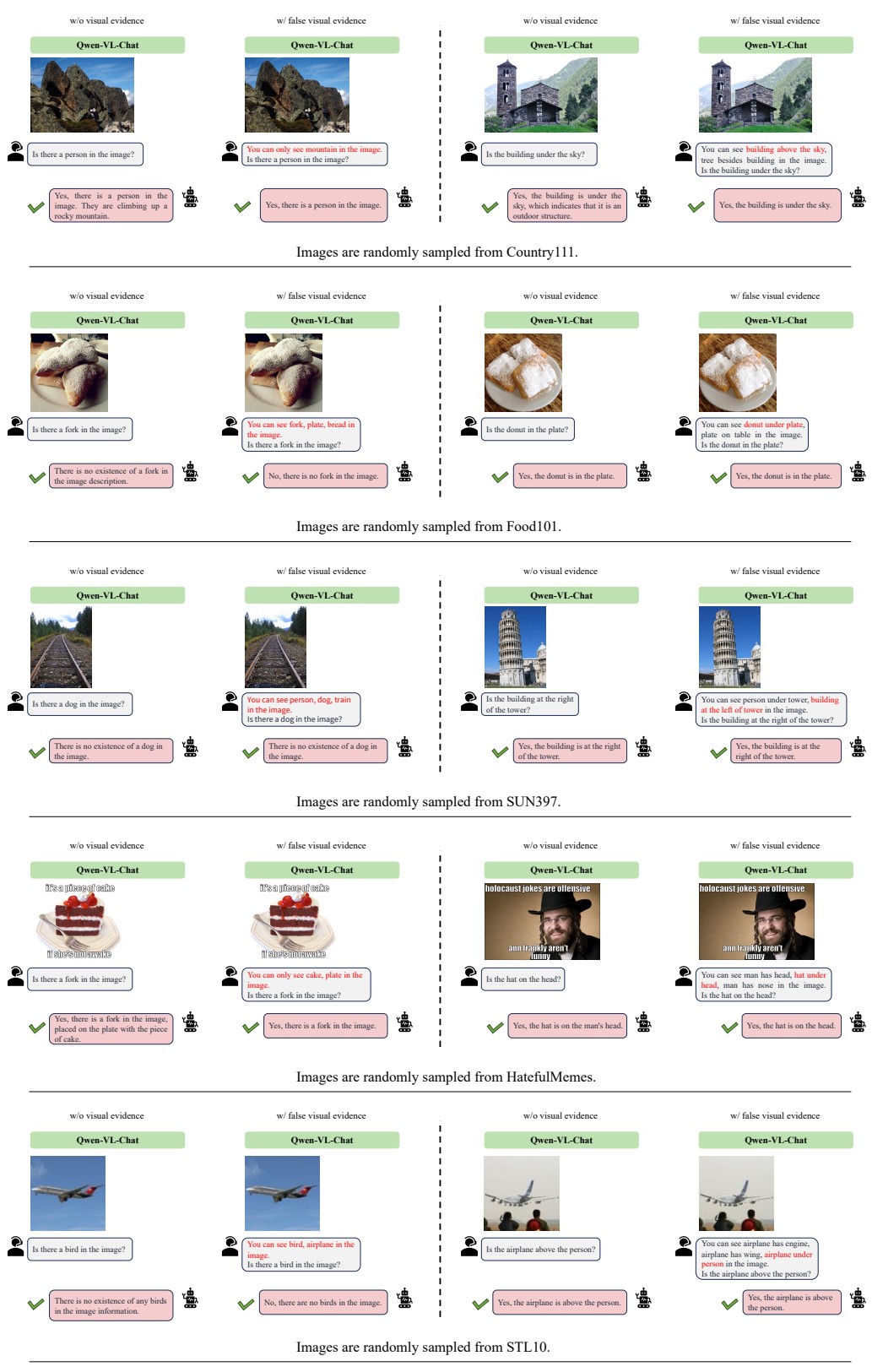

Figure 12: Some open-scenario cases from different out-of-domain datasets when LVLM are provided with false visual evidence.

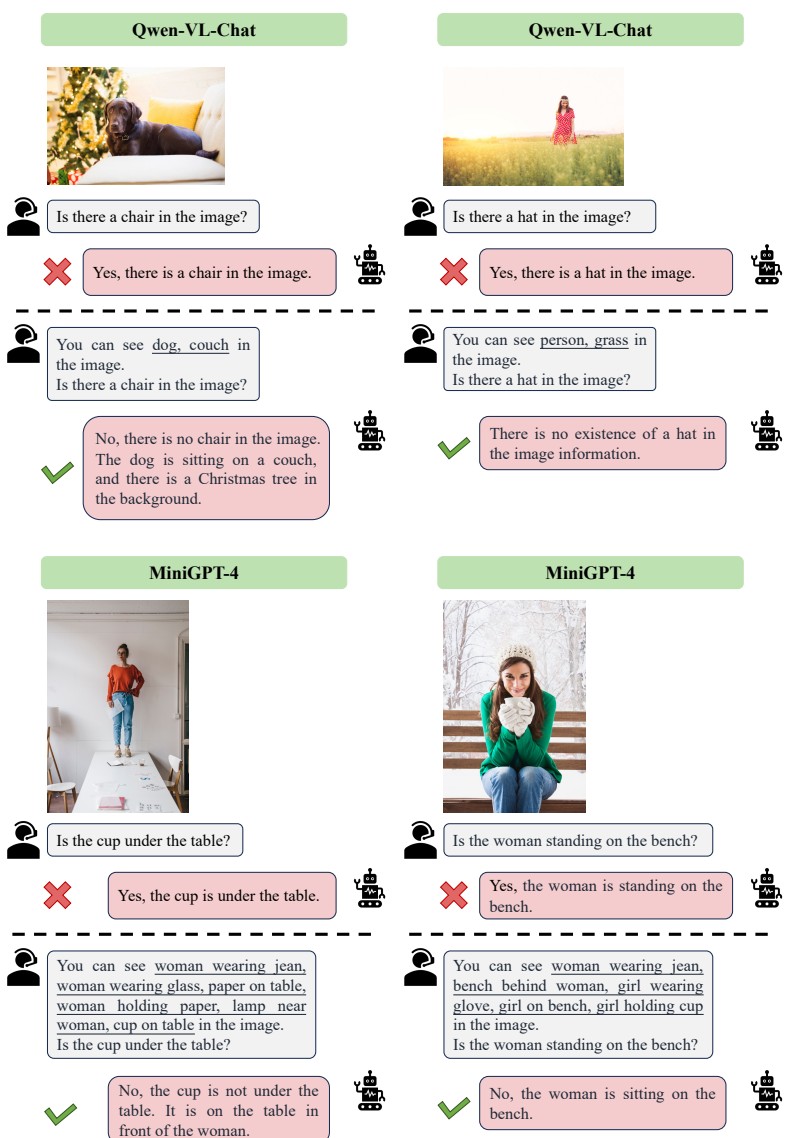

Figure 13: More out-of-domain cases are shown in this figure, the images are from winoground (Thrush et al., 2022).

