# OpenReview forum: "Visual Evidence Prompting Mitigates Hallucinations in Multimodal Large Language Models"
_ICLR.cc/2024/Conference — Submitted to ICLR 2024_

### Official Review · Reviewer_bJ8t · 2023-10-30

**Soundness:** 2 fair
**Presentation:** 2 fair
**Contribution:** 2 fair
**Rating:** 5
**Confidence:** 4

**Summary:**

This paper explores to mitigate hallucinations of LVLMs by a few visual knowledge evidence prompting provided as small visual models. Experiments on three large language models demonstrate its effectiveness on object hallucinations as well as relation hallucinations.

**Strengths:**

- well organized and easy to follow.
- a new idea to mitigate hallucinations of LVLMs.

**Weaknesses:**

- The authors are recommended to provide the performance comparisons on object and relation hallucinations where only object labels in the input image are provided.
If the performance is also good, please make explanations on the necessity of the proposed method.

- The authors are recommended to provide the hallucinations mitigations on open vocabulary objects and few-shot relations.

- make explanations on the decline of 'at' or slight improvement on 'under', displayed in Figure 4.

- make necessary analysis for situations where the performance of RelTR on SGG is significantly higher than motifs, while the performance of hallucination mitigations on LVLMs is equal or even opposite, displayed as in Table 9.

**Questions:**

As aforementioned

---

> ### Author Response · Authors · 2023-11-18
> **Response to reviewer bJ8t**
>
> We sincerely thank the reviewer for taking the time to review. We appreciate that you find our approach a new idea, well-organized and easy to follow. According to your valuable comments, we provide detailed feedback.
>
> **Q1:** Only providing object labels:
> > The authors are recommended to provide the performance comparisons on object and relation hallucinations where only object labels in the input image are provided. If the performance is also good, please make explanations on the necessity of the proposed method.
>
> ***Ans for Q1:***
>
> Thanks for the inspiring and interesting comments!
>
> We conduct validation experiments using the detr-resnet-101 model to provide object labels as evidence for relation hallucination.
>
> | Relation Hallucination | baseline | + object label |+ relation label |
> | --- | ---| --- | --- |
> | Qwen-VL-Chat | 63.62 | 71.41 | 75.68 |
> | mPLUG-Owl| 62.58 | 66.88 | 68.46 |
>
> The results show that providing object labels as evidence also has some improvement although not as effective as relation label. We suppose it is because object labels themselves contain crucial object information from the image, which leads to mitigating relation hallucination. This result not only validates the necessity of relation labels but also further verifies that our approach is orthogonal to the specific task. We have added the above discussions in the revision (Appendix A.4).
>
> **Q2:** Open-vocabulary objects and few-shot relations:
> > The authors are recommended to provide the hallucinations mitigations on open vocabulary objects and few-shot relations.
>
> ***Ans for Q2:***
>
> Thanks for your insightful question. We conduct experiments in response to your valuable advice. The following discussions are added in the revision (Appendix A.5).
>
> a) open vocabulary objects: We construct a new out-of-domain object hallucination dataset using the test sets from Object365 [A]. This dataset is divided into two parts. One part includes 80 objects that appear in COCO, while the other portion consists of objects that do not appear in COCO. The performance of these two parts are shown in the table below. It can be observed that there is a consistent improvement in performance for both in-domain and out-of-domain object categories.
> | Model |  | In-domain objects | Out-of-domain objects |
> | --- | --- | --- | --- |
> | mPLUG-owl | baseline | 58.68 | 48.45 |
> |  | **+ visual evidence** | **65.38** | **60.87** |
> | Qwen-VL-Chat | baseline | 74.64 | 67.87 |
> |  | **+ visual evidence** | **79.77** | **75.10** |
>
> b) few-shot relation: We chose the bottom-10 tail relations as defined in [C] of VG to construct a medium-sized relation hallucination dataset with 1006 samples. We used OpenPSG as the SGG model and conducted experiments on Qwen-VL-Chat and mPLUG-Owl. The experiment results are shown in the table below, and it can be seen that our framework still achieves significant improvements in few-shot relations.
>
> | Model | | Accuracy (%) |
> | --- | --- | --- |
> | mPLUG-Owl | baseline | 61.23 |
> |  | **+ visual evidence** | **67.89** |
> | Qwen-VL-Chat | baseline | 55.67 |
> |  | **+ visual evidence** | **68.63** |
>
> > [A] Objects365: A Large-scale, High-quality Dataset for Object Detection
> > [B] Simple Open-Vocabulary Object Detection with Vision Transformers
> > [C] Learning from the Scene and Borrowing from the Rich: Tackling the Long Tail in Scene Graph Generation
>
> **Q3:** Figure 4 explanation:
> > make explanations on the decline of 'at' or slight improvement on 'under', displayed in Figure 4.
>
> ***Ans for Q3:***
>
> Thanks for your careful review. We conducted statistical analysis on the samples of these two relations. It is observed that "at" is naturally ambiguous with "on", "next to", and "in", and "under" is ambiguous with "below" and "around". We will carefully review the collected data to minize the confusions.
>
> **Q4:** Analysis of Table 9 in Appendix:
> > make necessary analysis for situations where the performance of RelTR on SGG is signiìcantly higher than motifs, while the performance of hallucination mitigations on LVLMs is equal or even opposite, displayed as in Table 9.
>
> ***Ans for Q4:***
>
> Thank you for your meticulous review. In the Table 9, different scene graph generation models (RelTR, MOTIFS and OpenPSG) have comparable improvements on mPLUG-Owl and Qwen-VL-Chat. For example, RelTR achieves 5.92% and MOTIFS achieves 5.8% improvement on mPLUG-Owl. RelTR achieves 11.35% and MOTIFS achieves 12.55% improvement on Qwen-VL-Chat. The gains brought by different scene graph generation models to LVLM are within a stable range (saturated).
>
> | LVLM | Small model | mAP | Accuracy (%) |
> | --- | --- | --- | --- |
> | mPLUG-owl | baseline | - | 62.58 |
> |  | RelTR | 18.9 |  68.50 (+5.92) |
> |  | MOTIFS | 20.0 | 68.38 (+5.8) |
> |  | OpenPSG | 28.4 | 68.25 (5.67)|
> | Qwen-VL-Chat | baseline | - | 63.62 |
> |  | RelTR | 18.9 | 74.97 (+11.35) |
> |  | MOTIFS | 20.0 | 75.80 (+12.18) |
> |  | OpenPSG | 28.4 | 76.17 (+12.55) |

---

> ### Author Response · Authors · 2023-11-20
> **Sincerely looking forward to your feedbacks**
>
> Dear reviewer bJ8t,
>
> We sincerely apologize for inconveniencing you, but we have no other choice at the end of the discussion period. We sincerely hope you can understand.
>
> Thank you for taking the time to review our work. If you have any additional concerns or comments that we may have missed in our responses, we would be most grateful for any further feedback from you to help us further enhance our work. Your support or feedback are very important to us. We greatly appreciate your constructive comments and efforts.
>
> Best regards
>
> Authors of #3272

---

> ### Author Response · Authors · 2023-11-21
> **Sincerely looking forward to your feedbacks**
>
> Dear reviewer bJ8t,
>
> We sincerely apologize for inconveniencing you again, but we really have no other choice as the discussion period is drawing to a close (only 1 days left). We sincerely hope you can understand.
>
> Thank you for taking the time to review our work. If you have any additional concerns or comments that we may have missed in our responses, we would be most grateful for any further feedback from you to help us further enhance our work. Your support or feedback are very important to us. We greatly appreciate your constructive comments and efforts.
>
> Sorry again for inconveniencing you.
>
> Best regards
>
> Authors of #3272

---

> ### Author Response · Authors · 2023-11-22
> **Sincerely request for an opportunity to discuss with you**
>
> Dear reviewer bJ8t,
>
> Deep apologies for the repeated interruption. Sorry so much!
>
> We are truly honored that you have reviewed our paper and provided valuable constructive suggestions. Sincerely thanks for these constructive suggestions which greatly aid in our paper’s refinement, we have diligently addressed every concern and question you raised during the initial review. We genuinely hope our responses have resolved your concerns and provided satisfactory explanations. We sincerely appreciate your dedication and valuable time.
>
> To further improve our work, we sincerely hope for a valuable opportunity to discuss with you. It would be greatly appreciated if we could have a chance to hear your feedback. Your feedback is highly valuable to our paper and greatly contributes to the entire community.
>
> Very sorry for inconveniencing you again. Hope you have a good day!
>
>
> Best regards and many thanks,
>
> Authors of #3272

---

### Official Review · Reviewer_YNjN · 2023-10-31

**Soundness:** 3 good
**Presentation:** 3 good
**Contribution:** 3 good
**Rating:** 6
**Confidence:** 4

**Summary:**

This work explores visual evidence prompting and shows how small visual models complement the LVLMs by effectively extracting contextual information from images to generate precise answers. Experiments on both object and relation halluaciations shows the effectiveness of the proposed method.

**Strengths:**

- This paper proposes a simple method on mitigating LVLM's object and relation hallucination problem.
- The proposed method shows improvement for all the models and for both object and relation hallucinations.
- The authors also propose a dataset and benchmark for relation hallucinations.
- The authors also provide in-depth analysis on visual evidence prompting.

**Weaknesses:**

There are some unanswered questions regarding visual evidence, please refer to the **Questions** section.

**Questions:**

1. Have the authors explored questions regarding how overlapped between objects in questions and objects in evaluation datasets' questions?
1.1 What is the current stats on how overlapping between objects in questions and objects in evaluation datasets' questions?
1.2 What if visual evidence prompt contains objects that are not in the question? what if the prompt contains objects that are not in the questions exclusively?
1.3 What if the object names are switched to the synonyms to the objects appear in question?


2. Another question is regarding the pixel locations of detected objects? does it matter to change the scale of the image? for example, `cup <0, 284, 133, 424>'  to '<0 /width, 284 / height, 133 / width, 424 / height>'. or to '<0 /width * 2, 284 / height * 2, 133 / width * 2, 424 / height *2>'. I wonder to what extend LVLMs will differentiate object relations

---

> ### Author Response · Authors · 2023-11-18
> **Response to reviewer YNjN**
>
> We sincerely thank the reviewer for taking the time to review. We appreciate that you find our approach simple, experimental improvement, contributions of dataset and benchmark and the in-depth analysis. According to your valuable comments, we provide detailed feedback.
>
> **Q1:** Overlap between objects:
> > Have the authors explored questions regarding how overlapped between objects in questions and objects in evaluation datasets' questions?
> > 1.1 What is the current stats on how overlapping between objects in questions and objects in evaluation datasets' questions?
> > 1.2 What if visual evidence prompt contains objects that are not in the question? what if the prompt contains objects that are not in the questions exclusively?
> > 1.3 What if the object names are switched to the synonyms to the objects appear in question?
>
> ***Ans for Q1:***
>
> Thanks for the inspiring and interesting comments!
>
> a) We calculate the current stats of the overlap using detr-resnet-101. In the 3,000 visual evidence prompts, there are 298 prompts that contains object that are not in the question (Type A), and 1,202 prompts that contain objects that are not in the questions exclusively (Type B).
>
>
> |  | LVLM right->right | LVLM right->wrong | LVLM wrong->right | LVLM wrong->wrong |
> | -------- | -------- | -------- | -------- | -------- |
> | Type A | 139 (46.7%)     | 8 (2.7%)    | 110 (36.9%)     | 41  (13.8%)   |
> | Type B | 415 (34.5%)     | 22 (1.8%)    | 563 (46.8%)     | 202  (16.8%)   |
>
> b) Following Figure 3 in the draft, we calculate the stats of samples which were initally answered corrently/wrongly and answer correctly/wrongly after provided with Type A/B prompts. In the 298 Type A prompts, 110 of which (36.9%) allievates the hallucination of LVLM with detr-resnet-101 on Qwen-VL-Chat. In the 1,202 Type B prompts, 563 of which (46.8%) allievates the hallucination of LVLM.
>
> c) With the help of ChatGPT, we manually change the object appear in question to its synonyms respectively. The evaluation of object hallucination slightly decreases from 87.70% to 86.53% on Qwen-VL-Chat and from 78.38% to 71.54% on mPLUG-Owl, but there is still a non-trival improvement over the baseline especially on mPLUG-Owl, the results are shown in the table below.
> | Model | Setting | Accuracy (%) |
> | ---| --- | --- |
> | mPLUG-Owl | baseline | 57.29 |
> |  | before switching to synonyms | 78.38 |
> |  | after switching to synonyms | **71.54** |
> | Qwen-VL-Chat | baseline | 81.23 |
> |  | before switching to synonyms | 87.70 |
> |  | after switching to synonyms | **86.53** |
>
>
> In summary, with the LVLM's intrinsic robust ability to understand the output visual evidence of small models, visual evidence prompting is also robust in terms of the specific words used for objects in the given problem. We have add the above discussions in the revision (Appendix A.3).
>
> **Q2:** Scale of Image:
> > does it matter to change the scale of the image? for example, "cup <0, 284, 133, 424>" to "<0 /width, 284 / height, 133 / width, 424 / height>". or to "<0 /width * 2, 284 / height * 2, 133 / width * 2, 424 / height *2>". I wonder to what extend LVLMs will diëerentiate object relations
>
> ***Ans for Q2:***
>
> Thanks for your careful review. In Qwen-VL-Chat, for any given bounding box, a normalization process is applied (within the range [0, 1000)).
> For example, x_input = x_pixel / width * 1000, y_input = y_pixel / height * 1000. So the scale of the image does not differentiate any object relations. We clarify this setting in Sec. 4.1.3 in the revision.

---

> ### Author Response · Authors · 2023-11-20
> **Sincerely looking forward to your feedbacks**
>
> Dear reviewer YNjN,
>
> We sincerely apologize for inconveniencing you, but we have no other choice at the end of the discussion period. We sincerely hope you can understand.
>
> Thank you for taking the time to review our work. If you have any additional concerns or comments that we may have missed in our responses, we would be most grateful for any further feedback from you to help us further enhance our work. Your support or feedback are very important to us. We greatly appreciate your constructive comments and efforts.
>
> Best regards
>
> Authors of #3272

---

> ### Author Response · Authors · 2023-11-21
> **Sincerely looking forward to your feedbacks**
>
> Dear reviewer YNjN,
>
> We sincerely apologize for inconveniencing you again, but we really have no other choice as the discussion period is drawing to a close (only 1 days left). We sincerely hope you can understand.
>
> Thank you for taking the time to review our work. If you have any additional concerns or comments that we may have missed in our responses, we would be most grateful for any further feedback from you to help us further enhance our work. Your support or feedback are very important to us. We greatly appreciate your constructive comments and efforts.
>
> Sorry again for inconveniencing you.
>
> Best regards
>
> Authors of #3272

---

> > ### Comment · Reviewer_YNjN · 2023-11-22
> >
> > Thanks authors for the response & clarification. The additional analysis and experiments on object overlap is indeed interesting and shows the proposed method still outperforms baselines. But the authors are encouraged to keep improving the method and making the approach truly generalizable.
> >
> > My rating is updated based on the provided response.

---

> > > ### Author Response · Authors · 2023-11-22
> > > **Sincerely thanks for your support and raising the score**
> > >
> > > Dear reviewer YNjN,
> > >
> > > Thanks for your prompt response despite such a busy period. We deeply appreciate your consideration in raising the score. Your valuable and constructive comments have greatly helped us in enhancing our work. We will try our best to keep improving our method.
> > >
> > > Sincerely wishing you a happy and fulfilling life, both personally and in your research pursuits. Thank you very much! Hope you have a good day!
> > >
> > > Best regards and many thanks,
> > >
> > > Authors of #3272

---

### Official Review · Reviewer_w1s8 · 2023-11-02

**Soundness:** 3 good
**Presentation:** 2 fair
**Contribution:** 3 good
**Rating:** 6
**Confidence:** 3

**Summary:**

The paper studies the object hallucinations in large-VLM. It points out that the vision model detects the explicit objects that can be used for visual prompts and mitigates the issue of object hallucination. Sufficient experiments shed light on how the vision model and the visual prompts affect object-level hallucination.

**Strengths:**

The paper has good motivation, i.e., the vision model affects the hallucinations and the detection results can be used as visual prompting for LLM. The experiments also demonstrate the effectiveness of the proposed idea. The paper is also well-written.

**Weaknesses:**

The authors utilize small detection and scene graph models as evidence, potentially introducing domain-specific knowledge that skews the results. It's crucial to clarify the differences in training data between these small models and the evaluation data. Additionally, the paper should address whether the proposed method remains effective when applied to out-of-domain models.

The authors should extend their comparisons to include boosting and bagging methods, especially considering the paper primarily demonstrates performance improvements for yes/no VQA questions. This would provide a more robust validation of the proposed method.

Given that the small detection models used in the study are contrasted with large language models (LLMs), it would be beneficial to also present results using larger versions of these detection models. This would ensure a fair and comprehensive comparison.

The paper should incorporate a wider array of multilingual large language models (MLLMs), particularly those trained on Visual Genome (VG) data. A comparison with models like LLaVA 1.5, which achieves state-of-the-art performance with a minimal domain data, is essential. The authors should delve into why evidence-based methods might be superior to sampling fine-tuning techniques in this context.

**Questions:**

See the weakness

---

> ### Author Response · Authors · 2023-11-18
> **Response to reviewer w1s8 (part 1)**
>
> We sincerely thank the reviewer for taking the time to review. We appreciate that you find our approach motivation good, experimental effectiveness and well-written. According to your valuable comments, we provide detailed feedback.
>
> **Q1:** Out-of-domain data:
> > The authors utilize small detection and scene graph models as evidence, potentially introducing domain-speciìc knowledge that skews the results.
> > a) It's crucial to clarify the differences in training data between these small models and the evaluation data.
> > b) Additionally, the paper should address whether the proposed method remains effective when applied to out-of-domain models.
>
> ***Ans for Q1:***
>
> a) Thanks for pointing out this potentially confusing point. There is no overlap between the training data and the evaluation data. We are using the open-source small models trained on the COCO  or Visual Genome training set. The test set we have constructed is derived from the COCO validation set or Visual Genome test set. We have clarified this in the revision in Sec. 4.1.1.
>
> b) Thanks for your insightful question. We have incorporated this discussion into our revision in Appendix A.1. Specifcally, we collect 2,540 additional samples from open-world datasets and scenarios (Object365 [A] and OpenImage [B]) to further evaluate the genaliation ability of our method quantitatively.
>
> |  | Model | Setting | Accuracy (%) |
> | --- | --- | --- | --- |
> | Object Hallucination (2000 OOD samples)| mPLUG-Owl | baseline | 52.04 |
> |  |  | **+ visual evidence** | **62.46** |
> |  | Qwen-VL-Chat | baseline | 70.25 |
> |  |  | **+ visual evidence** | **76.74** |
> | Relation Hallucination (540 OOD samples) | mPLUG-Owl | baseline | 58.52 |
> |  |  | **+ visual evidence** | **72.41** |
> |  | Qwen-VL-Chat | baseline | 73.93 |
> |  |  | **+ visual evidence** | **75.98** |
>
> The above results present the comparison with baseline results for the evaluation on out-of-domain datasets. After incorporating visual evidence prompting, all models enables more precise discernment of object or relation presence within the image.
>
> Besides Figure 5 in the draft that showing some out-of-domain cases, we also follow CLIP and select 2 samples from 10 open-world datasets (without groundtruth) for qualititive analysis, including CLEVER and Caltech 101. These 20 cases are in the revised Appendix F.1.
>
> The contribution of our method lies in combining small and large models, utilizing the domain-specific knowledge of small models to complement the large models. In practical applications, it is possible to customize different small models to tailor different domain knowledge. The Figure 3 in the draft verified that if the small models struggle to effciently capture visual evidence, incorrect evidence has limited impact on cases that are already correct.
>
> > [A] Objects365: A Large-scale, High-quality Dataset for Object Detection
> > [B] The Open Images Dataset V4: Unified image classification, object detection, and visual relationship detection at scale
>
> **Q2:** Boosting and bagging methods:
> > The authors should extend their comparisons to include boosting and bagging methods, especially considering the paper primarily demonstrates performance improvements for yes/no VQA questions. This would provide a more robust validation of the proposed method.
>
> ***Ans for Q2:***
>
> Thanks for your insightful comments. In response to the suggestion, we conduct the experiments of utilizing the conventional bagging methods [A] between LVLM and VQA models which does a plurality vote when predicting. Combining the Qwen-VL-Chat and N2NMNs, the evaluation accuracy of object and relation hallucination slightly increases from 81.23% to 83.02% and from 63.62% to 67.91%. This result further validates the effectiveness of our framework.
>
> | | baseline | bagging | Visual Evidence Prompting |
> | --- | --- | --- | --- |
> | Object Hallucination | 81.23 | **83.02** | **87.70** |
> | Relation Hallucination | 63.62 | **67.91** | **75.68** |
>
> > [A] Bagging predictors

---

> > ### Author Response · Authors · 2023-11-18
> > **Response to reviewer w1s8 (part 2)**
> >
> > **Q3:** Larger detection models:
> > > Given that the small detection models used in the study are contrasted with large language models (LLMs), it would be beneìcial to also present results using larger versions of these detection models. This would ensure a fair and comprehensive comparison.
> >
> > ***Ans for Q3:***
> >
> > Thanks for your valuable suggestions. We conduct experiments with larger open-source detection models, DINO [A], which is the top-tier model with 58.0 mAP in COCO 2017 val (detr-resnet-101 has 43.5 mAP).
> >
> > | LVLM | small model | mAP | Accuracy (%) |
> > | --- | --- | --- | --- |
> > | mPLUG-owl | baseline | - | 57.29 |
> > |  | yolos-tiny | 28.7 | 70.23 (+12.94) |
> > |  | yolos-small | 36.1 | 73.44 (+3.21) |
> > |  | detr-resnet-50 | 42.0 | 76.55 (+3.11)|
> > |  | detr-resnet-101 | 43.5 | 78.38 (+1.83)|
> > |  | DINO-4scale-swin | 58.0 | **79.44 (+1.06)** |
> > | Qwen-VL-Chat | baseline | - | 81.23 |
> > |  | yolos-tiny | 28.7 | 83.73 (+2.50) |
> > |  | yolos-small | 36.1 | 85.47 (+2.01) |
> > |  | detr-resnet-50 | 42.0 | 87.10 (+1.63)|
> > |  | detr-resnet-101 | 43.5 | 87.70 (+0.60)|
> > |  | DINO-4scale-swin | 58.0 | **89.17 (+1.46)** |
> >
> > From the results, it can be observed that as the mAP increases, the small model consistently provides a boost to the large model, though it gradually saturates.
> >
> > **Q4:** More LVLMs:
> > > The paper should incorporate a wider array of multilingual large language models (MLLMs), particularly those trained on Visual Genome (VG) data. A comparison with models like LLaVA 1.5, which achieves state-of-the-art performance with a minimal domain data, is essential.
> >
> > ***Ans for Q4:***
> >
> > Thanks for your constructive comments. Firstly, we would like clarify that Qwen-VL-Chat is already trained on Visual Genome. And following your valuable advice, we conduct more experiments on LLaVA and LLaVA-1.5.
> >
> > |  | Model | Setting | Accuracy (%) |
> > | --- | --- | --- | --- |
> > | Object Hallucination | LLaVA-1.5 | baseline | 84.47 |
> > |  |  | **+ visual evidence** | **90.20** |
> > |  | LLaVA | baseline | 60.23 |
> > |  |  | **+ visual evidence** | **77.43** |
> > | Relation Hallucination | LLaVA-1.5 | baseline | 70.38 |
> > |  |  | **+ visual evidence** | **75.08** |
> > |  | LLaVA | baseline | 64.49 |
> > |  |  | **+ visual evidence** | **70.54** |
> >
> > It is observed that the hallucination evaluation of LLaVA-1.5 is indeed state-of-the-art, with an accuracy of 84.47% for object hallucination. However, it still exhibits a significant amount of relation hallucination, with an accuracy of 70.38%. Besides LLaVA, visual evidence prompting further helps LLaVA-1.5 alleviate both object and relation hallucination capabilities 84.47% -> 90.20%, 70.38% -> 75.08%, thereby providing further validation of the effectiveness of our method. This indicates that not only can different small models help alleviate hallucinations in large models, but a single small model can consistently alleviate hallucinations in large models of different sizes and trained on different datasets. This result further confirms the complementarity between large and small models and the necessity of our framework. We add these discussions in Appendix D.2 of our revision.
> >
> > **Q5:** Compared with finetuning techniques:
> > > The authors should delve into why evidence-based methods might be superior to sampling fine-tuning techniques in this context.
> >
> > ***Ans for Q5:***
> >
> > Thanks for your constructive suggestion. It is known that the foundation models gain speciality to achieve exceptional performance on the fine-tuning task, but it can potentially **lose its generality [B], i.e., catastrophic forgetting [D]**.
> > Previous work [C] has conducted fine-tuning experiments on LLaVA. As the fine-tuning progresses, LLaVA starts to **hallucinate** by disregarding the questions and exclusively generating text based on the examples in the fine-tuning datasets. As in the Table 3 in [C], after 1 epoch finetuning LLaVA-7b on MNIST, the accuracy on CIFAR-10 significantly drops from 56.71% to 9.26%. In response to your suggestion, we also validate the 7b-linear-ft-miniImagenet model in [C] in the object hallucination benchmark. The accuracy is 53.90%, which 6.33% lower than the baseline 60.23%.
> > On the other hand, our prompt-based method does **not modify the parameters of the model, and offer greater controllability**, which is advantageous for **preserving the model's original generalization capability**. We have add the above discussions in the revision (Appendix A.2).
> >
> > > [A] DINO: DETR with Improved DeNoising Anchor Boxes for End-to-End Object Detection
> > > [B] Speciality vs Generality: An Empirical Study on Catastrophic Forgetting in Fine-tuning Foundation Models
> > > [C] Investigating the Catastrophic Forgetting in Multimodal Large Language Models
> > > [D] Why there are complementary learning systems in the hippocampus and neocortex: insights from the successes and failures of connectionist models of learning and memory

---

> ### Author Response · Authors · 2023-11-20
> **Sincerely looking forward to your feedbacks**
>
> Dear reviewer w1s8,
>
> We sincerely apologize for inconveniencing you, but we have no other choice at the end of the discussion period. We sincerely hope you can understand.
>
> Thank you for taking the time to review our work. If you have any additional concerns or comments that we may have missed in our responses, we would be most grateful for any further feedback from you to help us further enhance our work. Your support or feedback are very important to us. We greatly appreciate your constructive comments and efforts.
>
> Best regards
>
> Authors of #3272

---

> > ### Comment · Reviewer_w1s8 · 2023-11-20
> > **Response**
> >
> > Dear authors,
> >
> > Thanks for your detailed reply. My concerns are addressed and raise my score.
> >
> > Best regards

---

> > > ### Author Response · Authors · 2023-11-21
> > > **Sincerely thanks for your support and raising the score**
> > >
> > > Dear reviewer w1s8,
> > >
> > > Thanks for your swift reply despite such a busy period. We are very glad to hear that your concerns have been addressed in our response, and we sincerely appreciate that you can raise the score. Your valuable and constructive comments have greatly helped us in enhancing our work.
> > >
> > > Sincerely wishing you a happy and fulfilling life, both personally and in your research pursuits. Thank you!
> > >
> > > Best regards and thanks,
> > >
> > > Authors of #3272

---

### Official Review · Reviewer_rr7K · 2023-11-03

**Soundness:** 2 fair
**Presentation:** 3 good
**Contribution:** 2 fair
**Rating:** 5
**Confidence:** 4

**Summary:**

The research addresses the persistent issue of hallucination in Large Vision-Language Models (LVLMs, such as GPT-3) where these models tend to make predictions of objects and relations that do not exist in the input images. The study highlights that while traditional small visual models produce professional and accurate outputs, they lack the ability to effectively interact with humans.

The primary focus of the research is to investigate how small visual models can complement LVLMs by extracting contextual information from images to generate precise answers. The proposed approach, known as "visual evidence prompting," demonstrates a natural mitigation of hallucination in LVLMs. This technique involves providing visual knowledge as context to reduce the hallucination problem. The study includes experiments conducted on three large language models, showing improved performance in addressing object hallucinations and a new benchmark for relation hallucinations.

Overall, the research aims to serve as a baseline for challenging hallucination benchmarks and emphasizes the significance of exploring and analyzing the substantial visual evidence concealed within small visual models before fine-tuning LVLMs. The paper presents an intriguing approach to mitigating hallucination in LVLMs, but there may be room for further clarification and validation of the proposed method in different scenarios and datasets. Additionally, it would be beneficial to provide a more detailed discussion of potential applications and practical implications of the findings.

**Strengths:**

1. The motivation is clear
2. The presentation is good

**Weaknesses:**

In light of the issue of object and relation hallucinations within the existing LVLM model, this paper employs the object and relation information captured by the visual component of the mini model as a contextual query to mitigate the LVLM hallucination problem. However, this approach gives rise to two significant concerns:

Efficiency: What degree of efficiency degradation can be expected with the introduction of new mini models for object detection and scene graph generation into the original method?

Domain Compatibility: Is there any overlap between the datasets used for training the small target detection and scene graph generation models and the LVLM hallucination test set?
In the context of an open-world scenario where the small model might struggle to efficiently capture target and relation information, to what extent will the acquisition of false evidence affect the LVLM's otherwise accurate responses? It's worth noting that despite the findings presented in Figure 3, my skepticism regarding this issue persists.

**Questions:**

na

---

> ### Author Response · Authors · 2023-11-18
> **Response to reviewer rr7K**
>
> We sincerely thank the reviewer for taking the time to review. We appreciate that you find our approach intriguing, the motivation clear, and the presentation good. According to your valuable comments, we provide detailed feedback.
>
> **Q1:** Efficiency:
> > What degree of efficiency degradation can be expected with the introduction of new mini models for object detection and scene graph generation into the original method?
>
> ***Ans for Q1:***
> Thanks for your constructive comments. We compute the average inference time of several small models on one A100 GPU.
>
> | yolos-small | detr-resnet-101 | RelTR | OpenPSG |
> | -------- | -------- | -------- | -------- |
> |0.193s     | 0.163s     | 0.394s     |0.218s|
>
> Meanwhile, the average time for one inference of Qwen-VL-Chat and mPLUG-Owl is:
>
> | Qwen-VL-Chat | mPLUG-Owl |
> | -------- | -------- |
> | 1.077s    | 1.157s     |
>
> After incorporating the small models, the relative increase in inference time is an average of 21.78%. Moreover, in multi-turn dialogue scenarios, the image is only uploaded once, regardless of how many questions are asked. The small model only needs to perform inference once, making the cost of inference relatively smaller.
>
> **Q2:** Domain Compatibility:
> > a) Is there any overlap between the datasets used for training the small target detection and scene graph generation models and the LVLM hallucination test set?
> > b) In the context of an open-world scenario where the small model might struggle to efficiently capture target and relation information, to what extent will the acquisition of false evidence affect the LVLM's otherwise accurate responses?
>
> ***Ans for Q2:***
>
> a) Thanks for pointing out this potentially confusing point. There is no overlap between the datasets used for training small models and the LVLM hallucination test sets. We are using the open-source small models trained on the COCO  or Visual Genome training set. The test set we have constructed is derived from the COCO validation set or Visual Genome test set. We have clarified this in the revision in Sec. 4.1.1.
>
> b) Thanks for your insightful question. We conduct more evaluations and incorporate the discussions into our revision in Appendix A.1.
> * Firstly, we collect 2,540 additional samples from open-world datasets and scenarios to further quantitatively evaluate the genaliation ability of our method. More details of these two out-of-domain datasets Object365 [A] and OpenImage [B]are added in the revision (Appendix A.1). In 20.6% of the images, small model captures incorrect or partial correct object or relation information. With these visual evidences, **only 8%** of the false evidence confuse the LVLM and change the response from collect to wrong.
> * Secondly, besides Figure 3, we follow CLIP and select 2 samples from 10 open-world datasets (without groundtruth) for qualititive analysis. These 20 cases are in the revised Appendix F.1. From these results and cases, it can be seen that in **open-world scenarios, incorrect evidence still has limited impact on cases that are already correct**.
> * Thirdly, The fifth prompt template in Table 3 in the draft shows that we can tell the LVLM that *the evidence might be wrong and keep your answer if you think the evidence is wrong or missing.*
>
> In summary, quantitative and qualititive analysis show that false evidence has limited affect on LVLM's accurate responses with the help of carefully designed prompt.
>
> > [A] Objects365: A Large-scale, High-quality Dataset for Object Detection
> > [B] The Open Images Dataset V4: Unified image classification, object detection, and visual relationship detection at scale

---

> ### Author Response · Authors · 2023-11-20
> **Sincerely looking forward to your feedbacks**
>
> Dear reviewer rr7K,
>
> We sincerely apologize for inconveniencing you, but we have no other choice at the end of the discussion period. We sincerely hope you can understand.
>
> Thank you for taking the time to review our work. If you have any additional concerns or comments that we may have missed in our responses, we would be most grateful for any further feedback from you to help us further enhance our work. Your support or feedback are very important to us. We greatly appreciate your constructive comments and efforts.
>
> Best regards
>
> Authors of #3272

---

> ### Author Response · Authors · 2023-11-21
> **Sincerely looking forward to your feedbacks**
>
> Dear reviewer rr7K,
>
> We sincerely apologize for inconveniencing you again, but we really have no other choice as the discussion period is drawing to a close (only 1 days left). We sincerely hope you can understand.
>
> Thank you for taking the time to review our work. If you have any additional concerns or comments that we may have missed in our responses, we would be most grateful for any further feedback from you to help us further enhance our work. Your support or feedback are very important to us. We greatly appreciate your constructive comments and efforts.
>
> Sorry again for inconveniencing you.
>
> Best regards
>
> Authors of #3272

---

> ### Author Response · Authors · 2023-11-22
> **Sincerely request for an opportunity to discuss with you**
>
> Dear reviewer rr7K,
>
> Deep apologies for the repeated interruption. Sorry so much!
>
> We are truly honored that you have reviewed our paper and provided valuable constructive suggestions. Sincerely thanks for these constructive suggestions which greatly aid in our paper’s refinement, we have diligently addressed every concern and question you raised during the initial review. We genuinely hope our responses have resolved your concerns and provided satisfactory explanations. We sincerely appreciate your dedication and valuable time.
>
> To further improve our work, we sincerely hope for a valuable opportunity to discuss with you. It would be greatly appreciated if we could have a chance to hear your feedback. Your feedback is highly valuable to our paper and greatly contributes to the entire community.
>
> Very sorry for inconveniencing you again. Hope you have a good day!
>
>
> Best regards and many thanks,
>
> Authors of #3272

---

### Author Response · Authors · 2023-11-20
**Global Response**

We sincerely thank all reviewers for taking the time to review our work. We appreciate that you find that **our work is intriguing, motivation is clear, and the presentation is good** (Reviewer rr7K), **motivation is good, idea is effectiveness and paper is well-written** (Reviewer w1s8), **experimental improvement, contributions of dataset and benchmark and the in-depth analysis**  (Reviewer YNjN), **a new idea, well-organized and easy to follow** (Reviewer bJ8t).

Here, we provide an overview of the responses for the main questions about the out-of-domain concern.

**Q1: Is there any overlap between the datasets used for training small models and the LVLM hallucination test sets?** (Reviewer rr7K and w1s8)

***Overview of Ans for Q1:***
There is no overlap between the datasets used for training small models and the LVLM hallucination test sets. We are using the open-source small models trained on the COCO or Visual Genome training set. The test set we have constructed is derived from the COCO validation set or Visual Genome test set. We have clarified this in the revision in Sec. 4.1.1.

**Q2: out of domain compatibility** (Reviewer rr7K, w1s8, and bJ8t)

***Overview of Ans for Q2:***

a) We constructed two additional out-of-domain datasets to further quantitatively validate the generalization ability of our method for the mitigation of object hallucination and relation hallucination. We conducted experiments on these out-of-domain datasets, the experimental results indicate that **in out-of-domain open scenarios, incorporating visual evidence can still mitigate the hallucination of various LVLMs significantly**.

b) We would like to highlight that **the contribution of our **framework** lies in combining small and large models, utilizing the domain-specific knowledge of small models to complement the large models**. In practical applications, it is possible to customize different small models to tailor different domain knowledge. And in Appendix A.7, we also show some cases that other tasks such as object counting and OCR can also benefit from our framework, which showing the great potential in real-world application.

c) We also follow CLIP and randomly select 2 samples (one for object hallucination and another for relation hallucination) from 10 open-world out-of-domain datasets for qualitative analysis. These 20 cases are in the revised Appendix F.1. From the analysis in Figure 3 in the draft and these open-world cases, it can be seen that in open-world scenarios, **incorrect evidence has limited impact on cases that are already correct attributing to the robustness and strong generalization ability of large models in open scenarios**.

Thanks again for all the reviewers' valuable comments and efforts. In response to feedback, we provide individual detailed responses below to each reviewer, and we carefully updated the paper based on the reviewers’ suggestions (updates highlighted in blue). We greatly appreciate the reviewers for their time and feedback, and we hope that our responses and the revised manuscript adequately address all the concerns. Please let us know if you have further questions or concerns!

---

### Meta-Review · Area_Chair_XsKS · 2023-12-15

**Metareview:**

This paper introduces a novel method to reduce hallucinations in large vision-language models (LVLMs) by integrating small visual models for improved accuracy.
the paper does present several notable strengths. The concept of combining small and large models to mitigate hallucinations is novel. The authors provide a detailed experimental analysis, demonstrating the method's potential in certain contexts. Additionally, the introduction of new dataset and benchmark for evaluating relation hallucinations is a significant contribution to the field.
However, the work has some notable limitations. It raises concerns about the efficiency and scalability of integrating small models into LVLMs, especially in real-world applications. The method's effectiveness in open-world scenarios remains unclear, and there's a lack of robustness against false evidence. These issues lead to the recommendation of rejection despite the paper's promising aspects and contributions to the field. The authors are encouraged to address those concerns for a future revision.

**Justification For Why Not Higher Score:**

Same as above.

**Justification For Why Not Lower Score:**

N/A

---

### Decision · Program_Chairs · 2024-01-16

Reject